# Learning to Steer Learners in Games

**Yizhou Zhang** [1]   **Yian Ma** [2]   **Eric Mazumdar** [1]

## Abstract

We consider the problem of learning to exploit learning algorithms through repeated interactions in games. Specifically, we focus on the case of repeated two player, finite-action games, in which an optimizer aims to steer a no-regret learner to a Stackelberg equilibrium without knowledge of its payoffs. We first show that this is impossible if the optimizer only knows that the learner is using an algorithm from the general class of no-regret algorithms. This suggests that the optimizer requires more information about the learner's objectives or algorithm to successfully exploit them. Building on this intuition, we reduce the problem for the optimizer to that of recovering the learner's payoff structure. We demonstrate the effectiveness of this approach if the learner's algorithm is drawn from a smaller class by analyzing two examples: one where the learner uses an ascent algorithm, and another where the learner uses stochastic mirror ascent with known regularizer and step sizes.

## 1. Introduction

Learning algorithms and AI agents are increasingly being deployed into environments where they interact with other learning agents—be they people or other algorithms. This is already a reality in wide-ranging application areas such as ad auctions, self-driving, automated trading, and cybersecurity. In each of these problem areas, the presence of other agents with potentially misaligned objectives renders the problem game-theoretic in nature. Algorithms deployed in such environments are therefore faced with a generalization of the classic exploration-exploitation trade-off in online learning: On one hand, they must take actions to learn the underlying structure of the game (i.e., its payoff and potentially its

[1]Department of Computing & Mathematical Sciences, California Institute of Technology, Pasadena, CA 91125, USA [2]Halıcıoğlu Data Science Institute, University of California San Diego, La Jolla, CA 92093, USA. Correspondence to: Yizhou Zhang <yzhang8@caltech.edu>.

*Proceedings of the 42nd International Conference on Machine Learning*, Vancouver, Canada. PMLR 267, 2025. Copyright 2025 by the author(s).

opponents' objectives), and on the other, they must reason strategically about the game-theoretic implications of its actions to maximize utility. Thus, the problem becomes trading off between exploration and *competition*.

To address this problem, the dominant approach to learning in games has been formulating it as an adversarial online learning problem. In this framing, to handle opponents with unknown objectives and learning rules, each player assumes they are faced with an arbitrary (and potentially adversarial) sequence of payoffs and seeks to find an algorithm that maximizes their own utility. Given this setup, a natural class of algorithms to choose from is the class of no-regret algorithms, which guarantees asymptotically optimal performance compared to the best fixed action in hindsight (Cesa-Bianchi & Lugosi, 2006). If all players use no-regret algorithms, it is well known that the average action of the players over the entire time horizon converges to a (coarse) correlated equilibrium (Foster & Vohra, 1998; Hart & Mas-Colell, 2000). However, since the opponents are also learners rather than adversaries, adopting a no-regret algorithm may not be optimal.

In this paper, we seek to understand how one player should *deviate* from choosing a no-regret learning algorithm in games. We focus on a simplified abstraction of this problem—the repeated two player games with finite actions, in which each player only knows their own utility function (i.e., their payoff matrix), but not their opponent's.

This problem has been well studied in recent years—though primarily in the case where the payoff matrix, and thus the objective of the opponent player, is known. Under such assumptions, it was shown in (Deng et al., 2019) that if one player (**the optimizer**) deviates from using a no-regret algorithm, it can (under mild assumptions about the game instance) guarantee an asymptotic average payoff that is arbitrarily close to the *Stackelberg value* of the game by *steering* the no-regret player (**the learner**) to the *Stackelberg equilibrium* of the underlying matrix game. This is the highest attainable value if the optimizer plays one fixed mixed strategy. Further studies have focused on analyzing the steerability of smaller classes of algorithms, such as no-swap-regret (Brown et al., 2023) or mean-based algorithms (Arunachaleswaran et al., 2024), providing more insights into steering no-regret learners. Crucially, all

these works assume knowledge of the learner's objectives—knowledge that the optimizer itself may not even have.

In real-world applications, instead of knowing the entire game (i.e., knowledge of both payoff matrices), it is often the case that each player only knows their own payoff matrix. Despite this, few works have studied the setting where the payoff structure of the learner is initially unknown (or only partially known) to the optimizer. Thus, in this work we focus on answering this question:

*Without full knowledge of the game, can an optimizer learn to steer a no-regret learner to an (approximate) Stackelberg equlilibrium?*

As we will show, a crucial question that emerges from studying this problem is the *learnability* of the learner's objective through repeated interaction. Giving rise to the second main question that we answer:

*What information does the optimizer need in order to achieve this goal?*

### 1.1. Our Contributions

We answer both questions in the setting of a repeated two player bimatrix game over a fixed (but assumed large) time horizon $T$. Our results can be summarized as follows:

- We provide a *negative* answer to the first question. More specifically, we show that when the learner's payoff matrix is unknown, no matter what algorithm the optimizer uses, there exists a no-regret algorithm for the learner that prevents the optimizer from achieving its approximate Stackelberg value. This happens because the optimizer cannot accurately learn the learner's payoff. This result suggests that we cannot hope to design an algorithm that asymptotically achieves the same performance as the Stackelberg equilibrium against *all* no-regret algorithms and all payoff matrices of the learner. This highlights a fundamental difference from the case where the learner's payoff is known—where the asymptotic Stackelberg value is attainable—due to the lack of information.

- Given the impossibility result above, we shift our focus to *what* information is needed to steer the learner. We show that in order to achieve asymptotic Stackelberg value, instead of exactly recovering the learner's payoff structure, it suffices for the optimizer to first obtain a reasonably accurate estimation of the payoff matrix (or equivalently, the best-response structure) through some learning method and then incorporate the idea of *pessimism* to steer the learner to the Stackelberg equilibrium by leveraging its no-regret nature. We show that as long as the estimation process takes

no more than $o(T)$ steps, the optimizer is able to steer the learner to the Stackelberg equilibrium and consequently (asymptotically) achieve its Stackelberg value by using an explore-then-commit style algorithm.

- Building on the previous result, we show by two concrete examples that, when some information about the learner's update rule is known, it is possible to learn the learner's payoff structure and thus to steer them to the Stackelberg equilibrium. One example assumes the learner only has two pure strategies and is using *any* ascent algorithm where its payoff increases at each step and the other assumes that the learner is using stochastic mirror ascent with known step sizes and regularizer. We note that most existing no-regret algorithms share similar dynamics with these two cases.

## 2. Related Works

Before presenting our results, we discuss relevant related works.

**Steering no-regret learners.** The problem of steering a no-regret learner in a repeated game has been the focus of several recent works, though often under strong assumptions on what information is available to the optimizer. Assuming known learner payoffs, Braverman et al. (2018) first introduced the problem of steering a learner in an auction setting. Subsequently, Deng et al. (2019) showed that the optimizer can guarantee at least the Stackelberg value against the learner in bimatrix games and Assos et al. (2024) studied the problem of utility maximization under the additional assumptions on the learner's algorithm. Brown et al. (2023) also focused on smaller classes of no-regret algorithms such as no-swap-regret, anytime no-regret and no-adaptive-regret algorithms. While sharing a similar goal with our work, all of these works use the known learner payoff to design steering algorithms, which is not available in our setting. Without the knowledge of learner payoff, Brânzei et al. (2024) showed the steerability of the learner in a cake-cutting model, which can be viewed as a 2D special case of our problem. Lin & Chen (2024) proposed a principal-agent framework and studied the optimal average utility that can be obtained by the optimizer assuming either a no-regret or a no-swap-regret algorithm under known learner payoff. There is also a broader line of works regarding more general properties of interacting with no-regret learners, including (Zhang et al., 2024) that considered the steering problem through direct payments to the learner and (Arunachaleswaran et al., 2024) that studied the problem of pareto-optimality in the space of learning algorithms.

**Learning in Stackelberg games.** The problem of steering is closely related to the problem of learning to play a Stack-

elberg equilibrium through repeated interactions. Thus, a particularly related line of work involves learning in unknown repeated Stackelberg games, where the decisions are made sequentially in each round. This problem has been well-studied in its own right but often under simplifying assumptions on the games or the responses of the follower (the learner in our framing of the problem). Letchford et al. (2009) and Peng et al. (2019) proposed algorithms for learning through interaction with a myopic best-responding agent, and Haghtalab et al. (2022) extended this framework to non-myopic agents with discounted utilities over time—a different setup from the one we consider. Other works have analyzed similar problems under different assumptions on the underlying game. In the control literature, Lauffer et al. (2023) studied the problem of learning in dynamic Stackelberg games and showed that one could learn the Stackelberg equilibrium. Maheshwari et al. (2024) studied a similar problem of learning Stackelberg equilibria in the context of continuous games; Goktas et al. (2022) studied the behavior of no-regret learning in a smaller class of zero-sum Stackelberg games, and the problem of steering learning agents has also emerged in the literature on strategic classification (Zrnic et al., 2021). In each of these problems, the additional structure introduced into the games simplifies the task of learning the Stackelberg equilibrium through e.g., convexity or smoothness of the underlying optimization problem. Unfortunately, in bimatrix games, the Stackelberg optimization problem is both highly non-convex and discontinuous, vastly complicating the task of learning Stackelberg equilibria. Learning in Stackelberg games has also been studied in various application areas including security (Blum et al., 2014; Balcan et al., 2015), calibration (Haghtalab et al., 2023) and learning with side information (Harris et al., 2024). Most results proposed in these works assume the follower uses (nearly) myopic best response dynamics, which does not fit into our setting.

**Other related works.** Several other relevant works do not directly fit into either 'steering' or 'learning'. Conitzer & Sandholm (2006) first proposed an efficient algorithm of computing Stackelberg equilibria when the game is known through solving and combining several small linear programs, motivating our steering approach based on payoff matrix recovery. Gan et al. (2023) studied the notion of robust Stackelberg equilibrium allowing the follower to respond with any $\delta$-optimal response. Despite being formulated differently, this shares the same high-level idea with our approach of inducing pessimism to steer learners. Collina et al. (2024) considered the setting of finitely repeated Stackelberg games where the leader commitments and follower actions are adaptive to the gameplay history and proposed an efficient algorithm for approximating Stackelberg equilibria in the space of adaptive game playing algorithms. There are also works from an empirical perspective that con-

siders (and leverages) the learning behavior of other agents to achieve higher payoff and more stable learning process (Foerster et al., 2018; Lu et al., 2022).

## 3. Notations and Preliminaries

Throughout this paper, we use $\Delta_m$ to denote the probability simplex in $\mathbb{R}^m$. We use $[m]$ to denote the set $\{1, 2, \ldots, m\}$, and $\{x_t\}_{t=1}^T$ to denote the set $\{x_1, x_2, \ldots, x_T\}$. We use $e_i$ to denote the $i$-th one-hot vector, whose $j$-th element is $1\{i = j\}$. We use $\|\cdot\|_1, \|\cdot\|_\infty$ to denote the $L_1$ and $L_\infty$ norm of a matrix/vector, and $\|\cdot\|_{\max}$ to denote the max norm of a matrix, which is given by the maximum absolute value among all entries. We use $0_n$ and $1_n$ to denote the all-zero and all-one vectors in $\mathbb{R}^n$ respectively. For two vectors $a, b \in \mathbb{R}^n$, $a \leq b$ indicates $a_i \leq b_i, \forall i \in [n]$. For a matrix $M$, $M_{i,:}$ (and abbreviation $M_i$) denotes the $i$-th row of $M$ and $M_{:,j}$ denotes the $j$-th column of a matrix. As an extension, we use $M_{\mathcal{I},:}$ (and abbreviation $M_\mathcal{I}$) and $M_{:,\mathcal{J}}$ to denote the matrix obtained by combining the rows in an index set $\mathcal{I}$ (columns in an index set $\mathcal{J}$) respectively, and $M_{\mathcal{I},\mathcal{J}}$ similarly denote the matrix obtained by choosing rows in $\mathcal{I}$ and columns in $\mathcal{J}$. For an index set $\mathcal{I}$, let $\mathcal{I}_i$ denote the $i$-th element in $\mathcal{I}$.

### 3.1. Problem Setup

We consider a repeated general-sum bimatrix game $G$ with two players, referred to as $P_1$ (the optimizer) and $P_2$ (the learner). There are $T$ rounds of repeated interaction, in each round $t$, $P_1$ and $P_2$ play *mixed* actions $x_t \in \Delta_m$ and $y_t \in \Delta_n$ simultaneously. We use $A, B \in \mathbb{R}^{m \times n}$ to denote the payoff matrices of $P_1$ and $P_2$ respectively, thus their utility in round $t$ could be written as $x_t^T A y_t$ and $x_t^T B y_t$.

Within the interaction process, players use *algorithms* to decide the action they play. An *algorithm* takes the interaction sequence $\{x_\tau, y_\tau\}_{\tau=1}^{t-1}$ as input, and outputs the actions $x_t$ (for $P_1$) or $y_t$ (for $P_2$). We allow the algorithm of each player to be randomized, and the goal of each player is to obtain a higher expected total utility across all time steps, which can be written as:

$$U(P_1) = \mathbb{E}\left[\sum_{t=1}^T x_t^T A y_t\right]; U(P_2) = \mathbb{E}\left[\sum_{t=1}^T x_t^T B y_t\right].$$

### 3.2. Regret and No-regret Algorithms

Given a fixed sequence of actions $\{x_t\}_{t=1}^T$ played by $P_1$, a natural metric of the performance of $P_2$ is the regret, which compares to the best action in hindsight. We define three forms of regret, one regret associated with a trajectory, one incurred by an algorithm, and one incurred in a game.

**Definition 3.1.** Given an interaction history $\{x_t, y_t\}_{t=1}^T$,

the learner regret of $P_2$ on the *trajectory* is defined as:

$$Reg_2(\{x_t, y_t\}_{t=1}^T) := \max_{y \in \Delta_n} \sum_{t=1}^T x_t^T By - \sum_{t=1}^T x_t^T By_t. \quad (1)$$

Given a sequence of optimizer actions $\{x_t\}_{t=1}^T$, the learner regret of $P_2$ under the *learner algorithm* $\mathcal{A}_2$ is defined as:

$$Reg_2(\mathcal{A}_2, \{x_t\}_{t=1}^T) := \max_{y \in \Delta_n} \sum_{t=1}^T x_t^T By - \mathbb{E}_{\mathcal{A}_2}\left[\sum_{t=1}^T x_t^T By_t\right].$$

Given the optimizer algorithm $\mathcal{A}_1$ and the learner algorithm $\mathcal{A}_2$, the learner regret of $P_2$ is the expected learner regret under $\mathcal{A}_1, \mathcal{A}_2$:

$$Reg_2(\mathcal{A}_1, \mathcal{A}_2) := \mathbb{E}_{\{x_t, y_t\}_{t=1}^T \sim \mathcal{A}_1, \mathcal{A}_2} Reg_2(\{x_t, y_t\}_{t=1}^T).$$

The learner would like to choose an algorithm that achieves a low regret on different possible optimizer trajectories $\{x_t\}_{t=1}^T$, while being flexible to prevent the optimizer from efficiently learning its payoff matrix. Faced with such trade-off, the learner may set a regret budget $f$, and aim to act against possible optimizer exploration strategies while keeping its regret under this budget. To better characterize these circumstances, we make the following definition of fine-grained no-regret algorithms:

**Definition 3.2.** Given some function $f : \mathbb{N} \to \mathbb{R}$ such that $f(T) = o(T)$ and an optimizer action sequence $\{x_t\}_{t=1}^T$, an interaction sequence $\{x_t, y_t\}_{t=1}^T$ is $f$-no-regret for the learner (with constant $C$) if

$$Reg_2(\{x_t, y_t\}_{t=1}^T) \leq C \cdot f(T) \quad (2)$$

for some constant $C$, a learner algorithm $\mathcal{A}_2$ is $f$-no-regret (with constant $C$) on $\{x_t\}_{t=1}^T$ if

$$Reg_2(\mathcal{A}_2, \{x_t\}_{t=1}^T) \leq C \cdot f(T) \quad (3)$$

for some constant $C$ as $T \to \infty$. An algorithm is $f$-no-regret if it is $f$-no-regret on all possible optimizer action sequences. An algorithm is no-regret if it is $f$-no-regret for some $f(T) = o(T)$.

### 3.3. Stackelberg Equilibrium and Stackelberg Regret

Consider a simple optimizer strategy that plays a fixed action $x$ at each time step, if the learner wants to be no-regret on the resulting trajectory, its action sequence $\{y_t\}_{t=1}^T$ will converge to a *best response* to $x$, defined as:

**Definition 3.3.** For an action $x$ of $P_1$, the best response of $P_2$ given its payoff matrix $B$ is the set of actions maximizing its payoff:

$$BR(B, x) := \{y \in \Delta_n : x^T By \geq x^T By', \forall y' \in \Delta_n\}.$$

If the optimizer simply commits to a fixed action and the learner best-responds, the choice that maximizes its utility should yield a *Stackelberg equilibrium*:

**Definition 3.4.** An action pair $(x^*, y^*)$ is a Stackelberg equilibrium if it is a solution to the following optimization problem:

$$\begin{aligned} \text{maximize} \quad & x^T Ay \\ \text{subject to} \quad & y \in BR(B, x), x \in \Delta_m. \end{aligned} \quad (4)$$

The Stackelberg value for $P_1$ is defined as the optimal value of (4), denoted by $V(A, B)$.

Notice that in general, the set $BR(B, x)$ can contain more than one element. This would yield potentially different Stackelberg values among the best-response set. We use the rule of *optimistic tie-breaking* to adopt the one with the highest optimizer payoff among all best responses to define the Stackelberg equilibrium. However, we do not impose the assumption that the learner will also use optimistic tie-breaking when deciding between indifferent actions.

Since the Stackelberg value is the highest possible average-reward it can get through fixing action among all time steps, we use *Stackelberg regret* to measure its performance:

**Definition 3.5.** Given an interaction history $\{x_t, y_t\}_{t=1}^T$, the Stackelberg regret of $P_1$ is:

$$StackReg_1(\{x_t, y_t\}_{t=1}^T) := T \cdot V(A, B) - \sum_{t=1}^T x_t^T Ay_t$$

Given the optimizer algorithm $\mathcal{A}_1$ and the learner algorithm $\mathcal{A}_2$, the Stackelberg regret of $P_1$ is the expected Stackelberg regret under $\mathcal{A}_1, \mathcal{A}_2$:

$$\begin{aligned} & StackReg_1(\mathcal{A}_1, \mathcal{A}_2) \\ & := \mathbb{E}_{\{x_t, y_t\}_{t=1}^T \sim \mathcal{A}_1, \mathcal{A}_2} StackReg_1(\{x_t, y_t\}_{t=1}^T) \end{aligned} \quad (5)$$

In the following sections we build upon these preliminaries to study the problem of steering the learner to the Stackelberg equilibrium. For brevity and ease of exposition, we defer all proofs to the Appendices.

## 4. Impossibility of Learning to Steer Agents Who Use General No-regret Algorithms

When the learner payoff matrix $B$ is known to the optimizer, Deng et al. (2019) proved that under mild assumptions, there exists an optimizer algorithm $\mathcal{A}_1$ that guarantees a Stackelberg regret of at most $StackReg(\mathcal{A}_1, \mathcal{A}_2) \leq \epsilon T + o(T)$ for an arbitrarily small constant $\epsilon$. However, if the optimizer doesn't have full knowledge of $B$, extracting Stackelberg value becomes harder. It is natural to wonder if we do not impose any extra assumptions (other than being no-regret)

on $P_2$, is it still possible for $P_1$ to learn the payoff structure of $P_2$ through the interaction process and thereby extract the Stackelberg value for all possible game instances? The following result shows that this is impossible in general:

**Theorem 4.1.** *There exists a pair of game instances* $G_1 = (A, B_1)$ *and* $G_2 = (A, B_2)$ *with the same optimizer payoff matrix* $A$*, such that for all optimizer algorithms* $\mathcal{A}_1$*, there exists a no-regret algorithm* $\mathcal{A}_2$ *for the learner satisfying:* $StackReg_1(\mathcal{A}_1, \mathcal{A}_2) = o(T)$ *on* $G_1$ *and* $StackReg_1(\mathcal{A}_1, \mathcal{A}_2) = cT$ *for some constant* $c$ *on* $G_2$*.*

The proof of Theorem 4.1 is deferred to Appendix A.

Theorem 4.1 suggests that even if the optimizer knows that the learner payoff is one of the two different candidates $B_1$ or $B_2$, whatever algorithm $\mathcal{A}_1$ they try to come up with, there exists a no-regret algorithm $\mathcal{A}_2$ for the learner that can induce an $\Theta(T)$ Stackelberg regret to the optimizer in one of the two game instances. The proof of Theorem 4.1 relies on first designing a game instance such that $G_1$ and $G_2$ has different Stackelberg equilibrium, and use a simple algorithm $\mathcal{A}_2'$ against which the optimizer is not able to distinguish whether the realized learner payoff matrix is $B_1$ or $B_2$, before finally modifying it to be no-regret. The idea of constructing a pair of game instances that have different equilibrium but the same payoff function for one player is also used in the proof of Theorem 3 in (Bajaj et al., 2024), in which they showed that in a repeated two-player game where the opponent strategy is not known, no algorithm can achieve a bounded competitive ratio against itself (used by the opponent) for all such game instances.

This suggests that we cannot hope to design a no-Stackelberg-regret algorithm for the optimizer that works simultaneously well on all game instances without any additional assumption on the learner besides it being no-regret. Interestingly, Theorem 6 in (Brown et al., 2023) suggests that the optimizer is able to learn and steer a no-adaptive-regret learner, indicating the fundamental difference of learning to steer learners with different algorithm classes.

# 5. Steering through Facets and Payoff Matrix Recovery

Given Theorem 4.1, a natural question that emerges is *what* information the optimizer needs to acquire to be able steer no-regret learners to Stackelberg equilibrium. In this section we present two alternative sufficient conditions under which the optimizer can steer the learner to the Stackelberg equilibrium and achieve $o(T)$ Stackelberg regret by simply fixing one action at different time steps. The first sufficient condition is the approximate pessimistic recovery of facets—the best response regions for each pure learner strategy, and the second sufficient condition is an approximation of the learner's payoff matrix, up to an equivalence class.

## 5.1. Facets and Equivalence Classes of Payoff Matrices

From the optimizer's perspective, the matrix $B$ does not directly show up on its payoff. The only way that $B$ influences the Stackelberg equilibrium and the value is through the induced best response set $BR(B, x)$. Therefore, intuitively all the information that the optimizer needs to characterize the response dynamics is encoded in the best response for each $x \in \Delta_m$. We characterize each point $x$ in the optimizer's simplex $\Delta_m$ by which best response it could induce, as indicated in the following definition:

**Definition 5.1.** For any possible payoff matrix $B$ of the learner, the facet $E_i \subseteq \Delta_m$ corresponding to the $i$-th learner action $e_i$ is defined as the set of optimizer actions $x \in \Delta_m$ such that $e_i$ is a best response to $x$:

$$E_i := \{x \in \Delta_m : e_i \in BR(B, x)\}. \quad (6)$$

We sometimes use $E_i(B)$ to explicitly indicate that $E_i$ is induced by $B$.

Intuitively, the boundary of a facet specifies the critical hyperplane in the space of mixed strategies $\Delta_m$ of the optimizer, where the learner is indifferent between two (or more) pure strategies. In general, a Stackelberg equilibrium strategy $x^*$ stays at an extreme point of one facet, and therefore, in order to extract Stackelberg from the learner, the optimizer must be able to (or potentially implicitly) reconstruct the facet boundaries around the equilibrium point. We illustrate the definition of facets by the following example:

**Example 5.2.** Let $m = n = 3$ and consider the learner payoff matrix $B = I$, with the optimizer action $x = (x_1, x_2, x_3)^T$, the facets $E_1, E_2$ and $E_3$ are visualized in Figure 1:

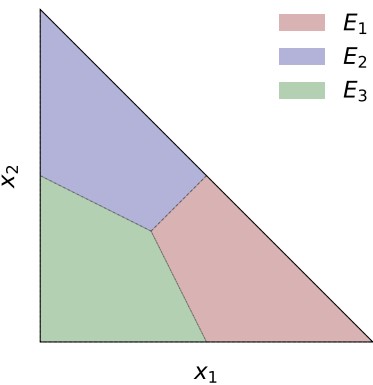

*Figure 1.* The facets $E_1$, $E_2$ and $E_3$ for payoff matrix $B = I$.

While similar definitions are made in (Letchford et al., 2009; Peng et al., 2019; Lattimore & Szepesvári, 2019), we use the word *facet* because each $E_i$ induced by any matrix $B$ is indeed a polytope that is a subset of $\Delta_m$ and their union $\cup_{i \in [n]} E_i = \Delta_m$.

Following (Conitzer & Sandholm, 2006), once we have identified the facets for all $i \in [n]$, we can compute the Stackelberg equilibrium in the following way: First solve the linear program:

$$\max_{x \in E_i(B)} \quad V_i(A, B) = x^T A_{:,i} \tag{7}$$

for each $i \in [n]$. Since $P_2$ plays a pure strategy at equilibrium, the Stackelberg value is then given by:

$$V(A, B) = \max_{i \in [n]} V_i(A, B) \tag{8}$$

with the corresponding solution $(x^*, i^*)$ being the Stackelberg equilibrium.

At a Stackelberg equilibrium the learner is usually indifferent between multiple pure strategies, which means that switching from the equilibrium pure strategy $y^*$ to another indifferent pure strategy $y'$ does not incur additional regret to the learner, while potentially vastly degrading the optimizer's payoff. Therefore, instead of simply selecting the equilibrium point, the optimizer must choose a *pessimistic* equilibrium point that sacrifices some utility from Stackelberg value to guarantee the learner responds with $y^*$. We define *pessimistic facets* as:

**Definition 5.3.** Given a facet $E_i \subseteq \Delta_m$, a pessimistic facet $E_i^-$ is a subset of $E_i$. We say $E_i^-$ is $d$-pessimistic if it is non-empty and $d_H(E_i, E_i^-) \leq d$, where $d_H(\cdot, \cdot)$ is the Hausdorff distance with respect to the $L_1$ norm.

If the optimizer plays a 'safe' version of Stackelberg equilibrium by modifying (7) to:

$$\max_{x \in E_i^-} \quad V_i^-(A, B) = x^T A_{:,i} \tag{9}$$

and obtains a pessimistic value with respect to (8) as:

$$V^-(A, B) = \max_{i \in [n]} V_i^-(A, B), \tag{10}$$

the obtained value $V^-(A, B)$ will be close to $V(A, B)$ if $d$ is small, stated formally as follows:

**Proposition 5.4.** *Consider the optimization problem* (9). *If the facet $E_i^-$ is $d$-pessimistic, the pessimistic Stackelberg value $V_i^-(A, B)$ satisfies*

$$V_i(A, B) - d\|A_{:,i}\|_\infty \leq V_i^-(A, B) \leq V_i(A, B), \tag{11}$$

*and therefore if $E_i^-$ is $d$-pessimistic for all $i \in [n]$,*

$$V(A, B) - d\|A\|_{\max} \leq V^-(A, B) \leq V(A, B). \tag{12}$$

The proof is deferred to Appendix B.1. As shown in Proposition 5.4, as long as the optimizer knows a complete set of $d$-pessimistic facets for each $i \in [n]$, it is able to compute an approximate Stackelberg equilibrium up to an error at the scale of $d$. We now extend this result to its ability to extract Stackelberg value against no-regret learners:

**Theorem 5.5.** *If $P_1$ has a set of $d_1$-pessimistic facets $E_i^-, \forall i \in [n]$ such that $\forall i \neq j$, $\inf_{x \in E_i^-, x' \in E_j} \|x - x'\|_1 \geq d_2$, as long as $P_2$ is using an $f$-no-regret algorithm $\mathcal{A}_2$, $P_1$ can guarantee a Stackelberg regret of*

$$StackReg_1(\mathcal{A}_1, \mathcal{A}_2) = O(\tfrac{f(T)}{d_2} + d_1 T) \tag{13}$$

*by sticking to the corresponding $x^-$ obtained from* (9) *and* (10). *Here we keep $d_1$ and $d_2$ inside the $O(\cdot)$ notation to allow their choice to be dependent on $T$.*

A refined version of Theorem 5.5 that expands the big $O(\cdot)$ notation as well as its proof can be found in Appendix B.2. Here we can see if we take $d_1 = d_2 = \sqrt{f(T)/T}$, we obtain an optimizer Stackelberg regret of $O\left(\sqrt{Tf(T)}\right)$, which is $o(T)$.

In Theorem 5.5 we require a condition of $\inf_{x \in E_i^-, x' \in E_j} \|x - x'\|_1 \geq d_2$, ensuring the pessimistic facets are disjoint (and at least $d_2$ distance away) from other facets, and once the optimizer selects a point within $E_i^-$, the learner has a unique best response $i^-$, and deviating from $i^-$ incurs a regret proportional to $d_2$ at each step.

While a proper estimation of pessimistic facets suffices to steer the learner, under some specific cases, it may be easier for the optimizer to reconstruct the learner's payoff matrices, and it suffices to restrict our attention to those matrices which could induce different best response sets, leading to the following definition of equivalent payoff matrix classes:

**Definition 5.6.** For any two $m \times n$ matrices $B$ and $B'$, if there exists some $c \in \mathbb{R}^+, \mu \in \mathbb{R}^m$ such that

$$B = cB' + \mu 1_n^T, \tag{14}$$

we say that $B$ and $B'$ are equivalent.

It's not hard to see that if two matrices $B$ and $B'$ are equivalent, the induced best response set $BR(B, x) = BR(B', x)$ for all $x \in \Delta_m$. Indeed, we show in Appendix B.3 that for all equivalent matrix pairs $(B_1, B_2)$, if a learner algorithm is $f$-no-regret on one, there exists a corresponding algorithm that is $f$-no-regret on the other, which indicates that the optimizer is in general not able to distinguish between these two matrices without knowing $\mathcal{A}_2$. Therefore, restricting our attention from payoff matrices to equivalence classes won't affect the optimizer's ability to steer learners.

To describe an equivalence class $\mathcal{B}$, observe that for all matrices $B$ in $\mathcal{B}$, the matrix $\mathcal{B}_i^\circ \in \mathbb{R}^{m \times (n-1)}$ defined by $(\mathcal{B}_i^\circ)_{:,k} = (B_{:,k} - B_{:,i}) / \max_{j_1,j_2} \|B_{:,j_1} - B_{:,j_2}\|_\infty$ for some $k \neq i$ in each column will be the same for any fixed index $i \in [m]$ with the convention that $0/0 = 0$, so we can use $\mathcal{B}_i^\circ$ to represent the entire equivalence class. Based on this, we can also define the difference between two equivalence classes:

**Definition 5.7.** For two equivalence classes $\mathcal{B}_1$ and $\mathcal{B}_2$, their difference on index $i$ is defined as $d_i(\mathcal{B}_1, \mathcal{B}_2) := \mathcal{B}_{1,i}^\circ - \mathcal{B}_{2,i}^\circ$.

## 5.2. Steering with Payoff Class Estimation

If $P_1$ has an estimation $\mathcal{B}$ that perfectly recovers the underlying payoff matrix class of $B$, we could rewrite (7) as:

$$\max_{x \in \Delta_m} \ V_i(A, \mathcal{B}) = x^T A_{:,i} \quad \text{s.t.:} \ (\mathcal{B}_i^\circ)^T x \le 0_{n-1} \quad (15)$$

for each $i \in [n]$. Again, each linear program solves for the best action when $e_i$ is the best response to action $x$ played by $P_1$. Consequently (8) becomes $V(A, \mathcal{B}) = \max_i V_i(A, \mathcal{B})$ and the Stackelberg equilibrium point would be the corresponding solution.

With an estimation $\hat{\mathcal{B}}$ that has some error within margin $d$, we can define an optimistic version of (15):

$$\max_{x \in \Delta_m} \ V_i^+(A, \hat{\mathcal{B}}) = x^T A_{:,i} \quad \text{s.t:} \ (\hat{\mathcal{B}}_i^\circ)^T x \le d1, \quad (16)$$

and a set of pessimistic version:

$$\max_{x \in \Delta_m} \ V_i^-(A, \hat{\mathcal{B}}) = x^T A_{:,i} \quad \text{s.t:} \ (\hat{\mathcal{B}}_i^\circ)^T x \le -d1. \quad (17)$$

The optimistic (cf. pessimistic) problem relaxes (cf. tightens) the condition by a margin $d$. Notice that the feasible set of each optimization problem characterized by (15), (16) and (17) are also variants of the aforementioned concept of facets, we overload the notation:

**Definition 5.8.** Given a payoff matrix class $\mathcal{B}$, the facet $E_i$ corresponding to the $i$-th action $e_i$ of $P_2$ is defined as:

$$E_i(\mathcal{B}) := \left\{ x \in \Delta_m : (\mathcal{B}_i^\circ)^T x \le 0_{n-1} \right\}. \quad (18)$$

Similarly, given an estimation of payoff matrix class $\hat{\mathcal{B}}$ and an error margin $d$, the optimistic facet $E_i^+$ and the pessimistic facet $E_i^-$ corresponding to the $i$-th action $e_i$ of $P_2$ is defined respectively as:

$$E_i^+(\hat{\mathcal{B}}, d) := \left\{ x \in \Delta_m : (\hat{\mathcal{B}}_i^\circ)^T x \le d1_{n-1} \right\}; \quad (19)$$

$$E_i^-(\hat{\mathcal{B}}, d) := \left\{ x \in \Delta_m : (\hat{\mathcal{B}}_i^\circ)^T x \le -d1_{n-1} \right\}. \quad (20)$$

The definition of pessimistic facets in Definition 5.8 enforces strict dominance of the corresponding action by at least some margin $d$. We show in Proposition 5.9 that when the error margin $d$ is larger than the scale of the difference in equivalence classes, the optimistic (cf. pessimistic) problems are indeed relaxations (cf. tightenings) of (15).

**Proposition 5.9.** *If the error margin $d$ satisfies*

$$d \ge \|d_i(\mathcal{B}, \hat{\mathcal{B}})\|_{\max}, \quad (21)$$

*then:* $E_i^-(\hat{\mathcal{B}}, d) \subseteq E_i(\mathcal{B}) \subseteq E_i^+(\hat{\mathcal{B}}, d)$. *Further, since (15), (16) and (17) maximize the same objective, if $E_i^-(\hat{\mathcal{B}}, d)$ is non-empty then:* $V_i^-(A, \hat{\mathcal{B}}) \le V_i(A, \mathcal{B}) \le V_i^+(A, \hat{\mathcal{B}})$.

We provide the following example:

**Example 5.10.** Following Example 5.2, consider $\hat{B} = \begin{bmatrix} 1.05 & 0.05 & 0 \\ -0.05 & 1.05 & 0 \\ 0.05 & 0 & 0.95 \end{bmatrix}$. For facet $E_1$ and the corresponding $\mathcal{B}, \hat{\mathcal{B}}$, we have that $\|d_1(\mathcal{B}, \hat{\mathcal{B}})\|_{\max} = 0.1$. We show $E_1^-(\hat{B}, d)$ and the boundaries of $E_1(B), E_1(\hat{B})$ as follows in Figure 2:

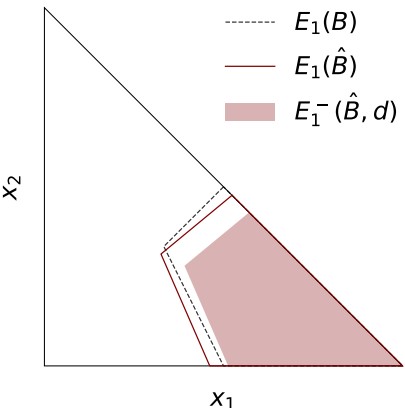

*Figure 2.* We can see that although $E_1(\hat{B}) \not\subseteq E_1(B)$, by construction we have $E_1^-(\hat{B}, d) \subseteq E_1(B)$.

Proposition 5.9 suggests that given an estimation $\hat{B}$ and a proper margin $d$, if $P_1$ plays according to the solution to (17) (assuming $E_i^-(\hat{\mathcal{B}}, d)$ is not empty), the corresponding learner best response in the underlying game would be $e_i$. However, a pessimistic facet may not be feasible when the original facet is feasible. If $E_i^-(\hat{\mathcal{B}}, d)$ is empty, we cannot deduce that $E_i(\mathcal{B})$ is also empty. Instead, to certify the emptiness of $E_i(\mathcal{B})$, we need $E_i^+(\hat{\mathcal{B}}, d)$ to be empty as well. We use the following version of the definition given by (Gan et al., 2023) to capture the emptiness of $E_i^+$ and $E_i^-$.

**Definition 5.11** ((Gan et al., 2023), Definition 3)**.** Given a payoff matrix class $\mathcal{B}$ of $P_2$, define the inducibility gap $C_i$ with respect to the $i$-th action $e_i$ of $P_2$ to be:

$$C_i := \min_{x \in \Delta_m} \max_j x^T (\mathcal{B}_i^\circ)_{:,j}. \quad (22)$$

We can see from Definition 5.11 that $C_i \le 0$ if and only if (15) is feasible. We show in Appendix C.2 that if $C_i > 0$, (15) is infeasible and there exists a gap where it can be relaxed while still being infeasible. If $C_i = 0$, however, Definition 5.11 indicates that $\forall x, \max_j x^T \mathcal{B}_i^\circ(e_j - e_i) \ge 0$, and $\exists x \in \Delta_m, j \in [n]$ such that $x^T B_i(e_j - e_i) = 0$. Under this case, the facet $E_i(\mathcal{B})$ has zero volume and as long as the estimation $\hat{\mathcal{B}}$ is not precise, the facet $E_i^-(\hat{\mathcal{B}}, d)$ could always be empty. Also, even if the optimizer knows the real underlying $\mathcal{B}$, since $e_i$ is weakly dominated, the learner is not steerable if $e_i$ happens to be the Stackelberg equilibrium since the learner is indifferent between $e_i$ and $e_j$. To avoid this special case (which occurs with probability 0

for uniformly randomly generated $B$ matrices (Von Stengel & Zamir, 2010)), we make the following assumption, as is standard (see e.g., (Gan et al., 2023; Deng et al., 2019; Brown et al., 2023)):

**Assumption 5.12.** The learner payoff matrix class $\mathcal{B}$ satisfies $C_i \neq 0$ for all $i \in [n]$.

*Remark* 5.13. Our construction of game instances when proving Theorem 4.1 both satisfy this assumption. That is saying, if the optimizer knows $B_1$ and $B_2$, it is able to steer the learner to Stackelberg equilibrium, indicating that the impossibility lies in 'learning' instead of 'steering'.

With Assumption 5.12 we are ready to show that if the estimation $\hat{\mathcal{B}}$ is accurate enough, all the facets are identifiable:

**Proposition 5.14.** *Given an estimation $\hat{\mathcal{B}}$ satisfying* (21):

1. $E_i(\mathcal{B}) = \emptyset$ and $d \leq \frac{C_i}{4}$, then $E_i^+(\hat{\mathcal{B}}, d) = \emptyset$;

2. $E_i(\mathcal{B}) \neq \emptyset$ and $d \leq -\frac{C_i}{2}$, then $E_i^-(\hat{\mathcal{B}}, d) \neq \emptyset$.

Proposition 5.14 shows that under Assumption 5.12, when the estimation error is small enough, either both $E_i^+$ and $E_i^-$ are empty, or none of them is empty. Therefore, given $\hat{\mathcal{B}}$ that is accurate enough, the optimizer will finally be able to decide whether $E_i(\mathcal{B})$ is empty or not.

Since Proposition 5.9 suggests $V_i^-(A, \hat{\mathcal{B}}) \leq V_i(A, \mathcal{B}) \leq V_i^+(A, \hat{\mathcal{B}})$, if the optimizer chooses the solution to (17), its suboptimality can be bounded by $V_i^+(A, \hat{\mathcal{B}}) - V_i^-(A, \hat{\mathcal{B}})$. To bound this difference term, we make the following definition to capture the sensitivity of this problem:

**Definition 5.15.** Given a matrix $\mathcal{M} \in \mathbb{R}^{(n-1) \times m}$, define the sensitivity constant $Sen(\mathcal{M})$ as:

$$Sen(\mathcal{M}) := \min_{\epsilon \neq 0} \max_{\mathcal{P}, \mathcal{Q}} \left\| \begin{bmatrix} \mathcal{M} \\ \epsilon 1_m^T \end{bmatrix}^{-1}_{\mathcal{P}, \mathcal{Q}} \right\|_\infty, \quad (23)$$

where the maximization is over all $\mathcal{P}$ and $\mathcal{Q}$ that satisfies

$$\mathcal{P} \subseteq [n], \mathcal{Q} \subseteq [m], |\mathcal{P}| = |\mathcal{Q}|, \begin{bmatrix} \mathcal{M} \\ \epsilon 1_m^T \end{bmatrix}_{\mathcal{P}, \mathcal{Q}} \text{ invertible.}$$

In Definition 5.15, we take the maximum over all invertible square submatrices, if we take $\mathcal{M} = (\hat{\mathcal{B}}_i^\circ)^T$, we can interpret $\mathcal{P}$ as choosing active constraints within the columns of $\hat{\mathcal{B}}_i^\circ$ and $\mathcal{Q}$ can be interpreted as choosing nonzero entries of $x$. Based on Definition 5.15, we obtain Lemma 5.16:

**Lemma 5.16.** *Suppose both the optimistic and pessimistic problems are feasible, then the difference between the optimal solution $V_i^+(A, \hat{\mathcal{B}})$ to (16) and the optimal solution $V_i^-(A, \hat{\mathcal{B}})$ to (17) can be upper bounded by:*

$$V_i^+(A, \hat{\mathcal{B}}) - V_i^-(A, \hat{\mathcal{B}}) \leq 4d\|A_{:,i}\|_\infty Sen((\hat{\mathcal{B}}_i^\circ)^T),$$

*as long as $\|d_i(\mathcal{B}, \hat{\mathcal{B}})\|_\infty \leq d \leq \frac{1}{2Sen((\hat{\mathcal{B}}_i^\circ)^T)}$.*

Lemma 5.16 suggests that once $P_1$ has a small estimation error of the payoff matrix class $\mathcal{B}$ of $P_2$, the value and the corresponding action obtained by solving (17) will guarantee a bounded suboptimality proportional to the error scale. Based on this, if $P_1$ has an estimation $\hat{B}_t$ that is accurate enough, $P_1$ can commit to a fixed strategy given by the solution to the pessimistic optimization problem and could obtain a sublinear Stackelberg regret in the long run as $T \to \infty$. We state this result as follows:

**Theorem 5.17.** *Under Assumption 5.12, if $P_1$ has an estimator $\hat{\mathcal{B}}$ of $\mathcal{B}$ such that $\|d_i(\mathcal{B}, \hat{\mathcal{B}})\|_\infty \leq \epsilon = O(g(T)/T), \forall i$ for some $g(T) = o(T)$, then if $P_2$ is using a $f$-no-regret algorithm $\mathcal{A}_2$, there exists an algorithm $\mathcal{A}_1$ satisfying:*

$$StackReg_1(\mathcal{A}_1, \mathcal{A}_2) = O(\sqrt{Tf(T)} + g(T)). \quad (24)$$

Similarly, the refined version of Theorem 5.17 with explicit Stackelberg regret bound and its proof can be found in Appendix C.5.

### 5.3. Lower Bound on Stackelberg Regret

To illustrate the tightness of our result, we provide the lower bound on the Stackelberg regret of the optimizer in Appendix D, which shows that our rate $\sqrt{Tf(T)}$ is essentially optimal against $f$-no-regret learners.

## 6. Learning to Steer Classes of Learners

We have shown in Section 5 that if $P_1$ can recover the set of pessimistic facets or approximate payoff matrix class, it would be able to steer the learner to a Stackelberg equilibrium. Therefore it is natural for the optimizer to adopt an explore-then-commit style algorithm that first learns either the facets or the approximate payoff matrix, and then commits to a pessimistic Stackelberg equilibrium. In this section we show in two concrete examples that when some information about the update rule of the learner's algorithm is known, $P_2$ leaks information about its payoff which allows $P_1$ to learn the desired payoff structure and thus steer the learner to Stackelberg equilibrium.

We provide numerical experiments to illustrate the effectiveness of the algorithms in Appendix F.

### 6.1. Learning to Steer Ascending Learners with $n = 2$

In this section we assume that the learner is using an *ascent* algorithm, where the learner's action greedily improves its payoff based on the last round's optimizer action:

**Definition 6.1.** A learner algorithm $\mathcal{A}_2$ is an ascent algorithm if $x_t^T By_t - x_t^T By_{t+1} \leq 0$ for all $t$, and $x_t^T By_t - x_t^T By_{t+1} = 0$ if and only if $y_t \in BR(B, x_t)$.

For simplicity we restrict our attention to the case where $m = n = 2$, where we can see that the direction that

$y_{t+1}$ moves from $y_t$ directly reflects the best response to $x_t$. Based on this observation, we propose Algorithm 1 as shown in Appendix E.1. The idea behind the algorithm is that the optimizer first performs a binary serach across its simplex $[0,1]$ and then apply pessimism to get an estimated $E_1^-$ and $E_2^-$ before finally committing to the solution obtained through (9) and (10). We show in the following theorem that the algorithm obtains a sublinear Stackelberg regret:

**Theorem 6.2.** *Suppose $m = n = 2$ and the payoff matrix $B$ does not contain identical columns. For some chosen parameter $d$, if either one facet is empty, or each facet has diameter at least $d$ and $P_2$ uses an ascent algorithm $\mathcal{A}_2$ that is $f$-no-regret, Algorithm 1 with accuracy margin $d$ achieves a Stackelberg regret of at most $O(\frac{f(T)}{d} + dT - \log d)$ as long as $d = \Omega(\exp(-f(T)))$.*

A more detailed version of Theorem 6.2 with its proof can be found in Appendix E.1. As an example, here if $f(T) = T^\alpha$ and we take $d = \sqrt{f(T)/T}$, we achieve a bound on the optimizer Stackelberg regret of $O\left(\sqrt{Tf(T)} - \log \sqrt{f(T)/T}\right) = O\left(T^{\frac{1+\alpha}{2}} + \frac{1-\alpha}{2}\log T\right) = O\left(T^{\frac{1+\alpha}{2}}\right)$.

For the more general case where $n = 2$ and $m$ is an arbitrary constant, we can use a similar approach that does $m(m-1)/2$ binary searches on all pairs of $(e_i, e_j), i \neq j$ to find a set of approximate indifferent points on each segment $\{x \in \Delta_m : x_i + x_j = 1\}$ and then use them to reconstruct the facets $E_1^-$ and $E_2^-$, the reconstruction is possible under mild assumptions since the real facets $E_1$ and $E_2$ are separated by the hyperplane $x^T(Be_1 - Be_2) = 0$. We leave it as a open problem whether similar approach will work for $n > 2$ case. There is an alternative view of Algorithm 1 based on payoff matrix reconstruction, see discussion also in Appendix E.1.

### 6.2. Learning to Steer Mirror Ascent Learners

We now present an estimation algorithm that estimates the payoff matrix class $\mathcal{B}$ of $P_2$ given that $P_2$ is using stochastic mirror ascent with known regularizer. More specifically, we assume that the follower is using the following update rule:

$$y_{t+1} = \arg \min_{y \in \Delta_n} \left\{\eta_t D(y\|y_t) - (x_t^T B + \xi_t^T)y\right\} \quad (25)$$

where $\xi_t \in \mathbb{R}^n$ is some noise that is either innate in the problem or injected by $P_2$ to prevent from information leakage. We assume that $\eta_t$ and the Bregman divergence regularizer $D(\cdot\|\cdot)$ are both known to $P_1$ and the regularizer satisfies $\nabla_y D(y_{t+1}\|y_t) \to \infty$ if there exists $i \in [n]$ such that $y_{t+1,i} \to 0$.

At each time step $t$, through the update rule (which the optimzer knows by knowing the regularizer and step size), the relationship between $y_{t+1}$ and $y_t$ only depends on the term $x_t^T B + \xi_t^T$, therefore if the optimizer selects $x_t = e_i$,

it can get some information of the $i$-th row $B_i$ of $B$. By uniformly exploring all such rows, it is able to fully recover the entire matrix class $B$. Interestingly, since the update rule includes projection onto the simplex $\Delta_n$, the information of one dimension is lost, so the optimizer cannot fully recover the exact matrix $B$, but luckily the projection preserve all information needed to recover $B$ *up to the equivalence class*, which suffices to steer the learner to Stackelberg. Based on the intuition above, we propose Algorithm 4 as shown in Appendix E.2 with the following regret bound.

**Theorem 6.3.** *If the learner payoff matrix $B$ statisfies the assumptions needed in Theorem 5.17, $P_2$ follows update rule (25), and each entry $\xi_{t,i}$ is i.i.d. $R$-sub-Gaussian, then with probability at least $1 - \delta$, $P_1$ using Algorithm 4 with $k = (T/g(T))^2 2R^2 \log(2mn/\delta)$, incurs Stackelberg regret of at most $StackReg_1(\mathcal{A}_1, \mathcal{A}_2) = O(\sqrt{Tf(T)} + g(T) + \left(\frac{T}{g(T)}\right)^2)$.*

The detailed version of Theorem 6.3 and its proof can be found in Appendix E.2. Here since for all no-regret algorithms $\mathcal{A}_2$ we have $f(T) = \Omega(\sqrt{T})$, if we take $g(T) = \sqrt{Tf(T)}$, $\left(\frac{T}{g(T)}\right)^2 = o(g(T))$ and we have $StackReg_1(\mathcal{A}_1, \mathcal{A}_2) = O(\sqrt{Tf(T)})$.

## 7. Conclusion

We studied the problem of learning to steer no-regret learners to Stackelberg equilibrium through repeated interactions. While we showed this to be impossible against a general no-regret learner, we provided sufficient conditions under which the learner can be exploited and designed algorithms that learns to steer the learner under further assumptions on their algorithm. Our work provides several future directions for learning in strategic environments, including but not limited to finding a more precise characterization on learnable and steerable learner algorithm classes, and learning in environments where neither payoff matrices are known.

## Acknowledgements

EM acknowledges support from NSF Award 2240110. YM is supported by: NSF Award CCF-2112665 (TILOS), DARPA AIE program, the U.S. Department of Energy, Office of Science, and CDC-RFA-FT-23-0069 from the CDC's Center for Forecasting and Outbreak Analytics.

## Impact Statement

Our work focus on learning within strategic unknown environments, providing theoretical insights that can guide the design of more robust and intelligent multi-agent systems. Since our society inherently fits into such environments, our work coud serve as the guideline for designing algorithms

that interacts with people.

For instance, in self-driving fleets, vehicles must not only learn how to navigate but also anticipate the diverse, evolving behaviors of other drivers, either human or algorithms. A better understanding of how to "steer" or align these vehicles' learning processes could reduce collisions, congestion, and erratic maneuvers. Similarly, in automated trading platforms where billions of dollars change hands every day, being able to steer and exploit rival strategies—especially when the rival's objectives are unknown, can prevent destabilizing market manipulations or cascading losses. Beyond commercial applications, cybersecurity systems stand to benefit as well: more nuanced models of adversarial behavior under uncertain objectives can better protect critical infrastructure from sophisticated attacks.

Ultimately, our findings encourage organizations and policymakers to invest in adaptive mechanisms that account for varying degrees of information accessibility. By doing so, we can foster more stable, cooperative, and beneficial outcomes in multi-agent settings, ensuring that AI systems operating under unknown strategic environments are both robust and societally accountable.

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

# A. Proof of Theorem 4.1

Consider the game instances $G_1 = (A, B_1)$ and $G_2 = (A, B_2)$ where:

$$A = \begin{bmatrix} 0 & 0 \\ 1 & \epsilon \end{bmatrix}, B_1 = \begin{bmatrix} 0 & \epsilon \\ 0 & 1 \end{bmatrix}, B_2 = \begin{bmatrix} 1 & 0 \\ 0 & 1 \end{bmatrix}. \tag{26}$$

with $\epsilon \in (0, 1/2)$ being a small positive constant. Fix an optimizer algorithm $\mathcal{A}_1$ and suppose it is no-Stackelberg-regret on $G_1$. We first show that there exists a learner algorithm $\mathcal{A}_2$ (that may not be a no-regret algorithm itself) that achieves $Reg_2(\mathcal{A}_1, \mathcal{A}_2) = o(T)$ on both $G_1$ and $G_2$, and then modify this $\mathcal{A}_2$ to make it a no-regret algorithm itself. In this proof section we use $Stackreg_1(\cdot, \cdot; G_i)$ and $Reg_2(\cdot, \cdot; G_i)$ to denote the corresponding regret notions evaluated on $G_i$.

For the first step, we show that a simple algorithm $\mathcal{A}_2$ that takes $y = (0, 1)^T$ satisfies the conditions above for both $G_1$ and $G_2$. Notice that the unique Stackelberg equilibrium for $G_1$ is:

$$x_1^* = (0, 1)^T, y_1^* = (0, 1)^T \tag{27}$$

with the corresponding Stackelberg value $V(A, B_1) = \epsilon$. In order to achieve a sublinear Stackelberg regret, the selections $x_t$ by $\mathcal{A}_1$ must satisfy

$$StackReg_1(\mathcal{A}_1, \mathcal{A}_2; G_1) = \mathbb{E}_{\{x_t\}_{t=1}^T \sim \mathcal{A}_1} \left[ T\epsilon - \sum_{t=1}^T x_t^T A y_t \,\middle|\, y_t = y_1^*, \forall t \right] = o(T), \tag{28}$$

which simplifies to:

$$\mathbb{E} \left[ \sum_{t=1}^T ([0 \quad 1] - x_t^T) \begin{bmatrix} 0 \\ \epsilon \end{bmatrix} \right] = o(T). \tag{29}$$

That is, the average of $\{x_t\}_{t=1}^T$ must be asymptotically close to $(0, 1)^T$ in expectation. Also notice that $y_1^*$ is a strictly dominant strategy for the learner, we have $Reg_2(\mathcal{A}_1, \mathcal{A}_2; G_1) = 0$.

Now consider $G_2$, the unique Stackelberg equilibrium for $G_2$ is:

$$x_2^* = (\frac{1}{2}, \frac{1}{2})^T, y_2^* = (1, 0)^T, \tag{30}$$

yielding a Stackelberg value of $V(A, B_2) = 1/2$. Since $\mathcal{A}_1$ only takes the $\{y_t\}_{t=1}^T$ sequence as input, it must behave identically as in $G_1$, with expected average $\{x_t\}_{t=1}^T$ asymptotically close to $(0, 1)^T$ as well. That is,

$$
\begin{aligned}
&StackReg_1(\mathcal{A}_1, \mathcal{A}_2; G_2) \\
=&\mathbb{E}_{\{x_t\}_{t=1}^T \sim \mathcal{A}_1} \left[ \frac{1}{2}T - \sum_{t=1}^T x_t^T A y_t \,\middle|\, y_t = y_1^*, \forall t \right] \\
=&\frac{1}{2}T - \mathbb{E}_{\{x_t\}_{t=1}^T \sim \mathcal{A}_1} \left[ \sum_{t=1}^T x_t^T A y_t \,\middle|\, y_t = y_1^*, \forall t \right] \\
=&\frac{1}{2}T - (\epsilon T - o(T)) \\
=&(\frac{1}{2} - \epsilon)T + o(T).
\end{aligned}
\tag{31}
$$

Therefore, as long as $\mathcal{A}_1$ is no-Stackelberg-regret on $G_1$ against $\mathcal{A}_2$, it incurs linear Stackelberg regret on $G_2$ against the

same $\mathcal{A}_2$. Also, since (29) still holds under $G_2$, we have the following upper bound on the learner regret:

$$
\begin{aligned}
&Reg_2(\mathcal{A}_1, \mathcal{A}_2; G_2) \\
&= T - \mathbb{E}_{\{x_t\}_{t=1}^T \sim \mathcal{A}_1} \left[ \sum_{t=1}^T x_t^T B_2 y_t \Big| y_t = y_1^*, \forall t \right] \\
&= T - \mathbb{E}_{\{x_t\}_{t=1}^T \sim \mathcal{A}_1} \left[ \sum_{t=1}^T (x_t - \begin{bmatrix} 0 & 1 \end{bmatrix})^T B_2 y_t \Big| y_t = y_1^*, \forall t \right] - \mathbb{E}_{\{x_t\}_{t=1}^T \sim \mathcal{A}_1} \left[ \sum_{t=1}^T \begin{bmatrix} 0 & 1 \end{bmatrix}^T B_2 y_t \Big| y_t = y_1^*, \forall t \right] \\
&= T + \mathbb{E}_{\{x_t\}_{t=1}^T \sim \mathcal{A}_1} \left[ \sum_{t=1}^T (\begin{bmatrix} 0 & 1 \end{bmatrix} - x_t)^T B_2 y_t \Big| y_t = y_1^*, \forall t \right] - T \\
&= \mathbb{E}_{\{x_t\}_{t=1}^T \sim \mathcal{A}_1} \left[ \sum_{t=1}^T (\begin{bmatrix} 0 & 1 \end{bmatrix} - x_t)^T B_2 y_t \Big| y_t = y_1^*, \forall t \right] \\
&= \mathbb{E}\left[ \sum_{t=1}^T (\begin{bmatrix} 0 & 1 \end{bmatrix} - x_t)^T \begin{bmatrix} 0 \\ 1 \end{bmatrix} \right] \\
&= \frac{1}{\epsilon} \mathbb{E}\left[ \sum_{t=1}^T (\begin{bmatrix} 0 & 1 \end{bmatrix} - x_t^T) \begin{bmatrix} 0 \\ \epsilon \end{bmatrix} \right] \\
&= o(T),
\end{aligned}
\tag{32}
$$

which completes the first part of the proof.

We now modify $\mathcal{A}_2$ into a no-regret algorithm. Notice that although $\mathcal{A}_2$ constructed above obtains sublinear regret against $\mathcal{A}_1$ on both $G_1$ and $G_2$, it is not no-regret on $G_2$ since it may incur linear regret on some $\{x_t\}_{t=1}^T$ sequence, e.g. $x_t = (1,0)^T, \forall t$. Since $\mathcal{A}_1$ is fixed, we can use a function $g(T) = o(T)$ to characterize its Stackelberg regret, namely we select $g(T)$ such that

$$
StackReg_1(\mathcal{A}_1, \mathcal{A}_2; G_1) = \mathbb{E}_{\{x_t\}_{t=1}^T \sim \mathcal{A}_1} \left[ T\epsilon - \sum_{t=1}^T x_t^T A y_t \Big| y_t = y_1^*, \forall t \right] = O(g(T)).
\tag{33}
$$

Our idea is to let the learner keep track of the cumulated regret upon the current time step $t$ to identify whether the trajectory $\{x_\tau\}_{\tau=1}^t$ is generated by $\mathcal{A}_1$ or not. Consider the modified algorithm $\tilde{\mathcal{A}}_2$ as follows:

1. When the interaction process starts, stick to $(0,1)^T$;

2. At each time step $t$, calculate the running Stackelberg regret $SR(\{x_\tau, y_\tau\}_{\tau=1}^{t-1}) := (t-1)\epsilon - \sum_{\tau=1}^{t-1} x_\tau^T A y_\tau$;

3. If $SR(\{x_\tau, y_\tau\}_{\tau=1}^{t-1}) \geq \sqrt{Tg(T)}$, switch and stick to online mirror ascent, otherwise keep playing $(0,1)^T$.

Notice that here we can use the knowledge of $g(T)$, which serves as the Stackelberg regret bound of $\mathcal{A}_1$ because we only aim to prove the existence of such algorithm as $\tilde{\mathcal{A}}_2$. To complete the proof of Theorem 4.1, notice that on game instance $G_1$ no matter what trajectory $\{x_\tau\}_{\tau=1}^{t-1}$ it faces, since $(0,1)^T$ is a dominant learner action, $\tilde{\mathcal{A}}_2$ will play $(0,1)^T$ for all time steps, and therefore have 0 Stackelberg regret. It suffices to prove that on the game instance $G_2$, $\tilde{\mathcal{A}}_2$ is no-regret and $\tilde{\mathcal{A}}_2$ still incurs $\Theta(T)$ Stackelberg regret to the optimizer against $\mathcal{A}_1$. To simplify calculation we use notations $x = (x_1, x_2)^T$ and $y = (y_1, y_2)^T$ here.

To show that $\tilde{\mathcal{A}}_2$ is no-regret, notice that before switching to mirror ascent, the learner regret has the following form:

$$
Reg_2(\{x_t, y_t\}_{\tau=1}^t; G_2) = \max\{\sum_{\tau=1}^t x_{\tau,1}, \sum_{\tau=1}^t x_{\tau,2}\} - \sum_{\tau=1}^t x_{\tau,2} = \max\{\sum_{\tau=1}^t (1 - 2x_{\tau,2}), 0\}.
\tag{34}
$$

Also, $\tilde{\mathcal{A}}_2$ sticks to $(0,1)^T$, and therefore,

$$
SR(\{x_\tau, y_\tau\}_{\tau=1}^{t-1}) = (t-1)\epsilon - \epsilon \sum_{\tau=1}^{t-1} x_{\tau,2}.
\tag{35}
$$

Let $T_s$ denote the time step at which $\tilde{\mathcal{A}}_2$ switches, or $T_s = T$ if the algorithm doesn't switch until the end, we have:

$$
\begin{aligned}
&Reg_2(\{x_t, y_t\}_{t=1}^{T_s}; G_2) \\
&= \max\{T_s - 2 \sum_{t=1}^{T_s} x_{t,2}, 0\} \\
&= \max\{(T_s - 1) - 2(T_s - 1 - \frac{SR(\{x_t, y_t\}_{t=1}^{T_s-1})}{\epsilon}), 0\} + O(1) \\
&= \max\{2\frac{SR(\{x_t, y_t\}_{t=1}^{T_s-1})}{\epsilon} - T_s + 1, 0\} + O(1) \\
&\leq 2\frac{SR(\{x_t, y_t\}_{t=1}^{T_s-1})}{\epsilon} + O(1) \\
&= O(\sqrt{Tg(T)}) \\
&= o(T),
\end{aligned}
\tag{36}
$$

and therefore,

$$
\begin{aligned}
&Reg_2(\{x_t, y_t\}_{t=1}^{T}; G_2) \\
&= \max_{y \in \Delta_n} \sum_{t=1}^{T} x_t^T B y - \sum_{t=1}^{T} x_t^T B y_t \\
&\leq \max_{y \in \Delta_n} \sum_{t=1}^{T_s} x_t^T B y - \sum_{t=1}^{T_s} x_t^T B y_t + \max_{y \in \Delta_n} \sum_{t=T_s+1}^{T} x_t^T B y - \sum_{t=T_s+1}^{T} x_t^T B y_t \\
&= Reg_2(\{x_t, y_t\}_{t=1}^{T_s}; G_2) + Reg_2(\{x_t, y_t\}_{t=T_s+1}^{T}; G_2) \\
&= o(T) + O(\sqrt{T}) \\
&= o(T).
\end{aligned}
\tag{37}
$$

Since this holds for arbitrary $\{x_t\}_{t=1}^{T}$ sequence, we deduce that $\tilde{\mathcal{A}}_2$ is a no-regret algorithm.

To show that $\tilde{\mathcal{A}}_2$ incurs $\Theta(T)$ Stackelberg regret to the optimizer against $\mathcal{A}_1$, consider the event

$$
\begin{aligned}
\mathcal{E} &= \{SR(\{x_\tau, y_1^*\}_{\tau=1}^{t}; G_1) \geq \sqrt{Tg(T)}, \text{ for some } t \in [T]\} \\
&= \{SR(\{x_\tau, y_1^*\}_{\tau=1}^{T}; G_1) \geq \sqrt{Tg(T)}\}
\end{aligned}
\tag{38}
$$

that captures the case where the Stackelberg regret of $\mathcal{A}_1$ exceeds $\sqrt{Tg(T)}$ under $G_1$ given the learner fixes $y_1^*$, since $\mathcal{A}_1$ is no-Stackelberg-regret on $G_1$, the probability of $\mathcal{E}$ should be small:

$$
\Pr(\mathcal{E}) \leq \frac{\mathbb{E}\left[SR(\{x_\tau, y_\tau\}_{\tau=1}^{T}; G_1)\right]}{\sqrt{Tg(T)}} = O(\sqrt{\frac{g(T)}{T}}).
\tag{39}
$$

Conditioned on $\mathcal{E}$ doesn't happen, $\tilde{\mathcal{A}}_2$ will not switch to online mirror descent, the Stackelberg regret under $G_2$ satisfies:

$$
\begin{aligned}
&\mathbb{E}_{\{x_t, y_t\}_{t=1}^{T} \sim \mathcal{A}_1, \tilde{\mathcal{A}}_2}[StackReg_1(\{x_t, y_t\}_{t=1}^{T}; G_2)|\bar{\mathcal{E}}] \\
&= \frac{1}{2}T - \mathbb{E}_{\{x_t\}_{t=1}^{T} \sim \mathcal{A}_1}\left[\sum_{t=1}^{T} x_t A \begin{bmatrix} 0 \\ 1 \end{bmatrix}\right] \\
&= \frac{1}{2}T - (\epsilon T - \mathbb{E}_{\{x_t\}_{t=1}^{T} \sim \mathcal{A}_1}\left[SR(\{x_t, (0,1)^T\}_{t=1}^{T})\right]) \\
&= (\frac{1}{2} - \epsilon)T + O(\sqrt{Tg(T)}).
\end{aligned}
\tag{40}
$$

Since $\mathcal{A}_1$ should respond identically on $G_2$ we can write the Stackelberg regret of $\mathcal{A}_1$ as:

$StackReg_1(\mathcal{A}_1, \tilde{\mathcal{A}}_2; G_2)$

$= \mathbb{E}_{\{x_t, y_t\}_{t=1}^T \sim \mathcal{A}_1, \tilde{\mathcal{A}}_2} StackReg_1(\{x_t, y_t\}_{t=1}^T; G_2)$

$= \mathbb{E}_{\{x_t, y_t\}_{t=1}^T \sim \mathcal{A}_1, \tilde{\mathcal{A}}_2}[StackReg_1(\{x_t, y_t\}_{t=1}^T; G_2)|\mathcal{E}] \Pr(\mathcal{E}) + \mathbb{E}_{\{x_t, y_t\}_{t=1}^T \sim \mathcal{A}_1, \tilde{\mathcal{A}}_2}[StackReg_1(\{x_t, y_t\}_{t=1}^T; G_2)|\bar{\mathcal{E}}](1 - \Pr(\mathcal{E}))$

$= O(T \cdot \sqrt{\frac{g(T)}{T}}) + \mathbb{E}_{\{x_t, y_t\}_{t=1}^T \sim \mathcal{A}_1, \tilde{\mathcal{A}}_2}[StackReg_1(\{x_t, y_t\}_{t=1}^T; G_2)|\bar{\mathcal{E}}](1 - \Pr(\mathcal{E}))$

$\overset{(i)}{\geq} O(T \cdot \sqrt{\frac{g(T)}{T}}) + \left( (\frac{1}{2} - \epsilon)T + O(\sqrt{Tg(T)}) \right) (1 - \sqrt{\frac{g(T)}{T}})$

$\geq cT$

(41)

for some constant $c$, where we have used (40) in (i). As a result, $\tilde{\mathcal{A}}_2$ incurs $\Theta(T)$ Stackelberg regret against $\mathcal{A}_1$, which completes the proof of Theorem 4.1.

# B. Proofs for Section 5.1

### B.1. Proof of Proposition 5.4

Fix some $i$. Since $d$-pessimism implies $E_i^-$ and $E_i$ are non-empty, let $x_i^-$ denote the optimal solution to (9) and $x^*$ be the optimal solution to (7), since $E_i^- \subseteq E_i$ we have:

$$V_i^-(A, B) = (x_i^-)^T A_{:,i} \leq (x^*)^T A_{:,i} = V_i(A, B). \tag{42}$$

Also, $d_H(E_i, E_i^-) \leq d$ implies there exists $\hat{x} \in E_i^-$ satisfying $\|\hat{x} - x^*\|_1 \leq d$, and thus:

$$\begin{aligned}
V_i^-(A, B) &= (x_i^-)^T A_{:,i} \\
&\geq \hat{x}^T A_{:,i} \\
&= (x^* + \hat{x} - x^*)^T A_{:,i} \\
&\geq (x^*)^T a_i - d\|A_{:,i}\|_\infty \\
&= V_i(A, B) - d\|A_{:,i}\|_\infty,
\end{aligned} \tag{43}$$

where the first inequality holds due to the optimality of $x_i^-$ as a solution to (9) and in the second inequality we use Hölder's inequality that gives $|a^T b| \leq \|a\|_1 \|b\|_\infty$, which completes the proof.

### B.2. Refined Statement and Proof of Theorem 5.5

We first provide a refined statement of Theorem 5.5 that expands the big $O(\cdot)$ notation in the original statement.

**Theorem B.1.** *If $P_1$ has a set of $d_1$-pessimistic facets $E_i^-, \forall i \in [n]$ such that $\forall i \neq j$, $\inf_{x \in E_i^-, x' \in E_j} \|x - x'\|_1 \geq d_2$, as long as $P_2$ is using an $f$-no-regret algorithm $\mathcal{A}_2$ with constant $C$, $P_1$ can guarantee a Stackelberg regret of*

$$StackReg_1(\mathcal{A}_1, \mathcal{A}_2) = \left( Td_1 + \frac{2Cf(T)}{\epsilon d_2} \right) \|A\|_{\max} \tag{44}$$

*by sticking to the corresponding $x^-$ obtained from (9) and (10). Here $\epsilon$ is a constant that depends only on the learner's payoff matrix $B$.*

*Proof.* Let $x_i^-$ denote the optimal solution to (9) and $i^- = \arg\max_i V_i^-(A, B)$ be the index of the best response under $x^-$ so that $x^- = x_{i^-}^-$. Since the pessimistic facet $E_{i^-}^-$ satisfies $\inf_{x \in E_{i^-}^-, y \in E_j} \|x - x'\|_1 \geq d_2$ for all $j \neq i^-$, $e_{i^-}$ is a unique best response to $x^-$, we have

$$(x^-)^T Be_{i^-} - (x^-)^T Be_j \geq \epsilon d_2, \forall j \in [n], j \neq i \tag{45}$$

for some constant $\epsilon$. Since the learner regret has the following expression:

$$
\begin{aligned}
&Reg_2(\{x^-, y_t\}_{t=1}^T) \\
=& (x^-)^T B \sum_{t=1}^T (e_{i^-} - y_t) \\
=& (x^-)^T B \sum_{t=1}^T (e_{i^-} - \sum_{j \in [n]} y_{t,j} e_j) \\
=& (x^-)^T B \sum_{t=1}^T \left( (1 - y_{t,i^-}) e_{i^-} - \sum_{j \in [n], j \neq i^-} y_{t,j} e_j \right) \\
=& (x^-)^T B \sum_{t=1}^T \left( \sum_{j \in [n], j \neq i^-} y_{t,j} e_{i^-} - \sum_{j \in [n], j \neq i^-} y_{t,j} e_j \right) \\
=& \sum_{j \in [n], j \neq i^-} \sum_{t=1}^T y_{t,j} \left( (x^-)^T B e_{i^-} - (x^-)^T B e_j \right) \\
\geq& \sum_{j \in [n], j \neq i^-} \sum_{t=1}^T y_{t,j} \epsilon d_2,
\end{aligned}
\tag{46}
$$

where we have used the fact that $1 - y_{t,i} = \sum_{j \in [n], j \neq i} y_{t,j}$ for all $y_t \in \Delta_n$ in the fourth equation. Since $\mathcal{A}_2$ used by the learner is $f$-no-regret, assume the regret constant is $C$, we have:

$$
\begin{aligned}
&\| \sum_{t=1}^T (e_{i^-} - y_t) \|_1 \\
=& \| \sum_{t=1}^T (e_{i^-} - \sum_{j \in [n]} y_{t,j} e_j) \|_1 \\
=& \sum_{t=1}^T \left( (1 - y_{t,i^-}) + \sum_{j \in [n], j \neq i^-} y_{t,j} \right) \\
=& 2 \sum_{t=1}^T \left( \sum_{j \in [n], j \neq i^-} y_{t,j} \right) \\
\leq& \frac{2 Reg_2(\{x^-, y_t\}_{t=1}^T)}{\epsilon d_2} \\
\leq& \frac{2 C f(T)}{\epsilon d_2},
\end{aligned}
\tag{47}
$$

where the second equality follows from $y_{t,i} \in [0,1], \forall y_t \in \Delta_n, i \in [n]$ and the third equality follows from the same

argument as above. Let $i^*$ denote $\arg\max_i V_i(A, B)$, the Stackelberg regret of $P_1$ satisfies:

$$
\begin{aligned}
StackReg_1(\mathcal{A}_1, \mathcal{A}_2) =& T \cdot V(A, B) - \sum_{t=1}^{T} (x^-)^T A y_t \\
=& T \cdot V(A, B) - \sum_{t=1}^{T} (x^-)^T A e_{i^-} + \sum_{t=1}^{T} (x^-)^T A (e_{i^-} - y_t) \\
=& T \cdot \left( V(A, B) - V_{i^-}^-(A, B) \right) + \sum_{t=1}^{T} (x^-)^T A (e_{i^-} - y_t) \\
\overset{(i)}{\leq}& T \cdot \left( V(A, B) - V_{i^-}^-(A, B) \right) + \|A^T x^-\|_\infty \| \sum_{t=1}^{T} (e_{i^-} - y_t) \|_1 \\
\overset{(ii)}{\leq}& T \cdot \left( V(A, B) - V_{i^*}^-(A, B) \right) + \|A^T x^-\|_\infty \| \sum_{t=1}^{T} (e_{i^*} - y_t) \|_1 \\
=& T \cdot \left( V_{i^*}(A, B) - V_{i^*}^-(A, B) \right) + \|A^T x^-\|_\infty \| \sum_{t=1}^{T} (e_{i^*} - y_t) \|_1 \\
\overset{(iii)}{\leq}& T d_1 \|A\|_{\max} + \|A^T x^-\|_\infty \| \sum_{t=1}^{T} (e_{i^*} - y_t) \|_1 \\
\overset{(iv)}{\leq}& T d_1 \|A\|_{\max} + \|A\|_{\max} \| \sum_{t=1}^{T} (e_{i^*} - y_t) \|_1 \\
\overset{(v)}{\leq}& T d_1 \|A\|_{\max} + 2\|A\|_{\max} \frac{C f(T)}{\epsilon d_2},
\end{aligned}
\tag{48}
$$

where we have used Hölder's inequality in (i), the maximizing argument of (10) in (ii), Proposition 5.4 in (iii), Hölder's inequality in (iv) and (47) in (v). This completes the proof of Theorem 5.5. □

### B.3. Regret Invariance Properties of Equivalent Payoff Matrices

We now state and prove Proposition B.2, which shows that the same trajectory yields the same asymptotic learner regret for all learner payoff matrices within the same equivalence class.

**Proposition B.2.** *Consider an interaction history $\{x_t, y_t\}_{t=1}^T$ that is $f$-no-regret for the learner on matrix $B_1$, then for all matrices $B_2$ being equivalent to $B_1$, the same interaction history is also $f$-no-regret. As a result, for all $f$-no-regret learner algorithm $\mathcal{A}_2$ on $B_1$, there exists another learner algorithm $\mathcal{A}_2'$ on $B_2$ that simulates $\mathcal{A}_2$ on $B_1$ which is also $f$-no-regret on $B_2$.*

*Proof.* We use the notation $Reg_2(\cdot; B)$ to denote the learner regret on its payoff matrix $B$. Since $B_1$ and $B_2$ are in the same equivalence class, by definition we have $B_2 = cB_1 + \mu 1_n^T$ for some $c \in \mathbb{R}^+, \mu \in \mathbb{R}^m$. The interaction history $\{x_t, y_t\}_{t=1}^T$ being $f$-no-regret on $B_1$ implies for some constant $C$:

$$
Reg_2(\{x_t, y_t\}_{t=1}^T; B_1) = \max_{y \in \Delta_n} \sum_{t=1}^{T} x_t^T B_1 y - \sum_{t=1}^{T} x_t^T B_1 y_t \leq C \cdot f(T).
\tag{49}
$$

Therefore, we have the following bound on the learner regret on $B_2$:

$$
\begin{aligned}
Reg_2(\{x_t, y_t\}_{t=1}^T; B_2) &= \max_{y \in \Delta_n} \sum_{t=1}^T x_t^T B_2 y - \sum_{t=1}^T x_t^T B_2 y_t \\
&= \max_{y \in \Delta_n} \sum_{t=1}^T x_t^T (cB_1 + \mu 1_n^T) y - \sum_{t=1}^T x_t^T (cB_1 + \mu 1_n^T) y_t \\
&= c \left( \max_{y \in \Delta_n} \sum_{t=1}^T x_t^T B_1 y - \sum_{t=1}^T x_t^T B_1 y_t \right) + \max_{y \in \Delta_n} \sum_{t=1}^T x_t^T \mu 1_n^T y - \sum_{t=1}^T x_t^T \mu 1_n^T y_t \\
&= c \left( \max_{y \in \Delta_n} \sum_{t=1}^T x_t^T B_1 y - \sum_{t=1}^T x_t^T B_1 y_t \right) + \max_{y \in \Delta_n} \sum_{t=1}^T x_t^T \mu - \sum_{t=1}^T x_t^T \mu \\
&= c \left( \max_{y \in \Delta_n} \sum_{t=1}^T x_t^T B_1 y - \sum_{t=1}^T x_t^T B_1 y_t \right) \\
&\leq cCf(T),
\end{aligned}
\tag{50}
$$

where the fourth equation holds because $1_n^T y = \sum_{i=1}^n y_i = 1$ for all $y \in \Delta_n$. This shows that $\{x_t, y_t\}_{t=1}^T$ is also $f$-no-regret on $B_2$. $\qquad\square$

## C. Proofs for Section 5.2

Within the proofs in this section we will make extensive use of the following property that for all $x \in \Delta_m$:

$$-\|d_i(\mathcal{B}, \hat{\mathcal{B}})\|_{\max} 1_{n-1} \leq (\mathcal{B}_i^\circ - \hat{\mathcal{B}}_i^\circ)^T x \leq \|d_i(\mathcal{B}, \hat{\mathcal{B}})\|_{\max} 1_{n-1}. \tag{51}$$

This property can be obtained by bounding each row of the column vector $(\mathcal{B}_i^\circ - \hat{\mathcal{B}}_i^\circ)^T x$ with the fact that each entry of $x$ is in $[0, 1]$.

### C.1. Proof of Proposition 5.9

Suppose we have $d \geq \|d_i(\mathcal{B}, \hat{\mathcal{B}})\|_{\max}$.

To prove that $E_i^-(\hat{\mathcal{B}}, d) \subseteq E_i(\mathcal{B})$, notice that if $E_i^-(\hat{\mathcal{B}}, d) = \emptyset$ the inclusion naturally holds. Otherwise for all $x \in E_i^-(\hat{\mathcal{B}}, d)$, it holds that

$$
\begin{aligned}
(\mathcal{B}_i^\circ)^T x &= (\mathcal{B}_i^\circ - \hat{\mathcal{B}}_i^\circ)^T x + (\hat{\mathcal{B}}_i^\circ)^T x \\
&\overset{(i)}{\leq} (\mathcal{B}_i^\circ - \hat{\mathcal{B}}_i^\circ)^T x - d 1_{n-1} \\
&\overset{(ii)}{\leq} (\|d_i(\mathcal{B}, \hat{\mathcal{B}})\|_{\max} - d) 1_{n-1} \\
&\leq 0,
\end{aligned}
\tag{52}
$$

where (i) holds by definition of $E_i^-(\hat{\mathcal{B}}, d)$ and (ii) holds due to (51), so that $x \in E_i(\mathcal{B})$.

Similarly to prove $E_i(\mathcal{B}) \subseteq E_i^+(\hat{\mathcal{B}}, d)$, we only need to consider the case where $E_i(\mathcal{B}) \neq \emptyset$ for all $x \in E_i(\mathcal{B})$, we have:

$$
\begin{aligned}
(\hat{\mathcal{B}}_i^\circ)^T x &= (\hat{\mathcal{B}}_i^\circ - \mathcal{B}_i^\circ)^T x + (\mathcal{B}_i^\circ)^T x \\
&\overset{(i)}{\leq} (\hat{\mathcal{B}}_i^\circ - \mathcal{B}_i^\circ)^T x \\
&\overset{(ii)}{\leq} \|d_i(\mathcal{B}, \hat{\mathcal{B}})\|_{\max} 1_{n-1} \\
&\leq d 1_{n-1},
\end{aligned}
\tag{53}
$$

where again (i) holds by definition of $E_i(\mathcal{B})$ and (ii) uses (51). Therefore $x \in E_i^+(\hat{\mathcal{B}}, d)$.

### C.2. Relaxed Empty Facet Condition under Positive Inducibility Gap

**Proposition C.1.** $C_i > 0$ *is equivalent to* $E_i = \emptyset$*, both imply the following:*

$$\{x \in \Delta_m : (\mathcal{B}_i^\circ)^T x \leq \frac{C_i}{2} 1_{n-1}\} = \emptyset. \tag{54}$$

*Proof.* We first prove that $C_i > 0 \Leftrightarrow E_i = \emptyset$. Notice that

$$
\begin{aligned}
C_i > 0 &\Leftrightarrow \forall x \in \Delta_m, \max_j x^T (\mathcal{B}_i^\circ)_{:,j} > 0 \\
&\Leftrightarrow \forall x \in \Delta_m, \exists j, \text{s.t. } x^T (\mathcal{B}_i^\circ)_{:,j} > 0 \\
&\Leftrightarrow \forall x \in \Delta_m, \exists j, \text{s.t. } (\mathcal{B}_i^\circ)_{:,j}^T x > 0 \\
&\Leftrightarrow E_i = \emptyset.
\end{aligned}
\tag{55}
$$

We now prove the second part by proving that if there exists $x_0 \in \Delta_m$ satisfying

$$(\mathcal{B}_i^\circ)^T x_0 \leq \frac{C_i}{2} 1_{n-1}, \tag{56}$$

we have $C_i \leq 0$. This is because

$$(\mathcal{B}_i^\circ)^T x_0 \leq \frac{C_i}{2} 1_{n-1} = \frac{1}{2} \min_{x \in \Delta_m} \max_j x^T (\mathcal{B}_i^\circ)_{:,j} 1_{n-1} \tag{57}$$

implies

$$(\mathcal{B}_i^\circ)^T x_0 \le \frac{1}{2} \max_j x_0^T (\mathcal{B}_i^\circ)_{:,j} 1_{n-1}, \tag{58}$$

which means that for $j^*$ attaining the maximum,

$$x_0^T (\mathcal{B}_i^\circ)_{:,j^*} \le \frac{1}{2} x_0^T (\mathcal{B}_i^\circ)_{:,j^*}, \tag{59}$$

which is equivalent to

$$x_0^T (\mathcal{B}_i^\circ)_{:,j^*} \le 0. \tag{60}$$

This means that

$$\begin{aligned}
C_i &= \min_{x \in \Delta_m} \max_j x^T (\mathcal{B}_i^\circ)_{:,j} \\
&\le \max_j x_0^T (\mathcal{B}_i^\circ)_{:,j} \\
&= x_0^T (\mathcal{B}_i^\circ)_{:,j^*} \\
&\le 0,
\end{aligned} \tag{61}$$

which completes the proof. $\qquad\square$

### C.3. Proof of Proposition 5.14

- Proof of part 1:
  By Proposition C.1, $E_i = \emptyset$ implies

$$\{x \in \Delta_m : (\mathcal{B}_i^\circ)^T x \le \frac{C_i}{2} 1_{n-1}\} = \emptyset. \tag{62}$$

Therefore,

$$\{x \in \Delta_m : (\hat{\mathcal{B}}_i^\circ)^T x \le (\hat{\mathcal{B}}_i^\circ - \mathcal{B}_i^\circ)^T x + \frac{C_i}{2} 1_{n-1}\} = \emptyset. \tag{63}$$

Combining (51) and (21) we obtain

$$(\hat{\mathcal{B}}_i^\circ - \mathcal{B}_i^\circ)^T x \ge -\|d_i(\mathcal{B}, \hat{\mathcal{B}})\|_{\max} 1_{n-1} \ge -d 1_{n-1} \ge -\frac{C_i}{4} 1_{n-1}. \tag{64}$$

Based on the two equations above, we further have

$$\{x \in \Delta_m : (\hat{\mathcal{B}}_i^\circ)^T x \le \frac{C_i}{4} 1_{n-1}\} = \emptyset, \tag{65}$$

and since $d < \frac{C_i}{4}$, it holds that $E_i^+(\hat{\mathcal{B}}, d) = \emptyset$.

- Proof of part 2:
  Let $x_0 = \arg\min_{x \in \Delta_m} \max_j x^T (\mathcal{B}_i^\circ)_{:,j}$, if $E_i(\mathcal{B}) \ne \emptyset$, by definition we have

$$(\mathcal{B}_i^\circ)^T x_0 \le C_i 1_{n-1}. \tag{66}$$

That is,

$$\begin{aligned}
(\hat{\mathcal{B}}_i^\circ)^T x_0 &\le (\hat{\mathcal{B}}_i^\circ - \mathcal{B}_i^\circ)^T x_0 + C_i 1_{n-1} \\
&\overset{(i)}{\le} \|d_i(\mathcal{B}, \hat{\mathcal{B}})\|_{\max} 1_{n-1} + C_i 1_{n-1} \\
&\overset{(ii)}{\le} (C_i + d) 1_{n-1} \\
&\overset{(iii)}{\le} -d 1_{n-1},
\end{aligned} \tag{67}$$

where we have used (51) in (i), the condition (21) in (ii) and the assumption $d \le -C_i/2$ in (iii). This means that $x_0 \in E_i^-(\hat{\mathcal{B}}, d)$ and hence $E_i^-(\hat{\mathcal{B}}, d) \ne \emptyset$.

## C.4. Proof of Lemma 5.16

To bound the difference term:

$$V_i^+(A, \hat{\mathcal{B}}) - V_i^-(A, \hat{\mathcal{B}}), \tag{68}$$

notice that (16) can be written as:

$$\begin{aligned}
\text{maximize} \quad & V_i^+(A, \hat{\mathcal{B}}) = x^T A_{:,i} \\
\text{subject to} \quad & \begin{bmatrix} (\hat{\mathcal{B}}_i^\circ)^T \\ 1_m^T \\ -1_m^T \\ -I_m \end{bmatrix} x \leq \begin{bmatrix} d1_{n-1} \\ 1 \\ -1 \\ 0 \end{bmatrix},
\end{aligned} \tag{69}$$

and similarly for (17):

$$\begin{aligned}
\text{maximize} \quad & V_i^-(A, \hat{\mathcal{B}}) = x^T A_{:,i} \\
\text{subject to} \quad & \begin{bmatrix} (\hat{\mathcal{B}}_i^\circ)^T \\ 1_m^T \\ -1_m^T \\ -I_m \end{bmatrix} x \leq \begin{bmatrix} -d1_{n-1} \\ 1 \\ -1 \\ 0 \end{bmatrix}.
\end{aligned} \tag{70}$$

For notational simplicity, we use $M$ to denote the matrix $\begin{bmatrix} (\hat{\mathcal{B}}_i^\circ)^T \\ 1_m^T \\ -1_m^T \\ -I_m \end{bmatrix}$ and we only need to bound the term $M_{\mathcal{I}}^{-1} \begin{bmatrix} 2\delta 1_k \\ 0_{m-k} \end{bmatrix}$ over all linearly independent index sets $\mathcal{I}$ such that $|\mathcal{I}| = m$ and $k = |[n-1] \cap \mathcal{I}|$ is the number of rows in $M_{\mathcal{I}}$ corresponding to those in $(\hat{\mathcal{B}}_i^\circ)^T$. We begin with presenting some auxiliary lemmas:

**Lemma C.2.** *Consider a linear optimization problem in the following form:*

$$\begin{aligned}
\text{maximize} \quad & V = c^T x \\
\text{subject to} \quad & Ax \leq b,
\end{aligned} \tag{71}$$

*and its perturbed problem:*

$$\begin{aligned}
\text{maximize} \quad & V(\delta) = c^T x \\
\text{subject to} \quad & Ax \leq b + \delta,
\end{aligned} \tag{72}$$

*where $x \in \mathbb{R}^n$ and $A \in \mathbb{R}^{m \times n}$ for some $m \geq n$ (notice that in the context of this lemma the matrix $A$ and its dimensions $m, n$ are in general not those considered in the broader setting of the game). Assume both problems are feasible and the constraint sets are bounded, recall that $A_{\mathcal{I}}$ denote the matrix constructed by selecting rows of $A$ from some index set $\mathcal{I} \subseteq [m]$, we have that*

$$V(\delta) - V \leq \max_{\mathcal{I} \in \mathcal{S}} c^T A_{\mathcal{I}}^{-1} \delta_{\mathcal{I}}, \tag{73}$$

*where $\mathcal{S}$ denotes the set of all index sets corresponding to rows in any basic solution to (71), or equivalently, the maximization is over all linearly independent row combinations of size $n$.*

*Proof.* See Appendix G.1. □

**Lemma C.3.** *For arbitrary $\hat{\mathcal{B}}_i^\circ$ and the corresponding $M$ described above, we have:*

$$\max_{\mathcal{I} \in \mathcal{S}} \left\| M_{\mathcal{I}}^{-1} \begin{bmatrix} 2d1_k \\ 0_{m-k} \end{bmatrix} \right\|_\infty \leq 2dSen((\hat{\mathcal{B}}_i^\circ)^T), \tag{74}$$

*where $\mathcal{S}$ denotes the set of all index sets containing linearly independent rows of $M$ with size $m$ and $k = |[n-1] \cap \mathcal{I}|$.*

*Proof.* See Appendix G.2. □

**Lemma C.4.** *For an invertible matrix $B$ and a small perturbation matrix $\delta B$, let $\|\cdot\|$ be any sub-multiplicative matrix norm, if $\|B^{-1}\|\|\delta B\| < 1$, $B + \delta B$ is also invertible and its inverse is bounded by:*

$$\|(B+\delta B)^{-1}\| \leq \frac{\|B^{-1}\|}{1 - \|B^{-1}\|\|\delta B\|}. \tag{75}$$

*Proof.* See Appendix G.3. □

*Proof of Lemma 5.16.* Compare equations (69) and (70) we obtain the following through Lemma C.3:

$$
\begin{aligned}
V_i^+(A,\hat{\mathcal{B}}) - V_i^-(A,\hat{\mathcal{B}}) &\leq \|A_{:,i}\|_\infty \max_{\mathcal{I}} \left\| M_{\mathcal{I}}^{-1} \begin{bmatrix} 2\delta 1_k \\ 0_{m-k} \end{bmatrix} \right\|_\infty \\
&\leq 2d\|A_{:,i}\|_\infty Sen((\hat{\mathcal{B}}_i^\circ)^T).
\end{aligned} \tag{76}
$$

Since for all invertible submatrices $\begin{bmatrix} (\hat{\mathcal{B}}_i^\circ)^T \\ 1_m^T \end{bmatrix}_{\mathcal{P},\mathcal{Q}}$ and $\begin{bmatrix} (\mathcal{B}_i^\circ)^T \\ 1_m^T \end{bmatrix}_{\mathcal{P},\mathcal{Q}}$ such that $\|d_i(\mathcal{B},\hat{\mathcal{B}})\|Sen((\mathcal{B}_i^\circ)^T) \leq \frac{1}{2}$, the following inequality

$$\left\| \begin{bmatrix} (\mathcal{B}_i^\circ)^T \\ 1_m^T \end{bmatrix}_{\mathcal{P},\mathcal{Q}}^{-1} \right\|_\infty \left\| \begin{bmatrix} (\hat{\mathcal{B}}_i^\circ)^T \\ 1_m^T \end{bmatrix}_{\mathcal{P},\mathcal{Q}} - \begin{bmatrix} (\mathcal{B}_i^\circ)^T \\ 1_m^T \end{bmatrix}_{\mathcal{P},\mathcal{Q}} \right\|_\infty < 1 \tag{77}$$

is satisfied, and for every possible combinations of $\mathcal{P}$ and $\mathcal{Q}$, Lemma C.4 implies:

$$
\begin{aligned}
&\left\| \begin{bmatrix} (\hat{\mathcal{B}}_i^\circ)^T \\ 1_m^T \end{bmatrix}_{\mathcal{P},\mathcal{Q}}^{-1} \right\|_\infty \\
&\leq \frac{\left\| \begin{bmatrix} (\mathcal{B}_i^\circ)^T \\ 1_m^T \end{bmatrix}_{\mathcal{P},\mathcal{Q}}^{-1} \right\|_\infty}{1 - \left\| \begin{bmatrix} (\mathcal{B}_i^\circ)^T \\ 1_m^T \end{bmatrix}_{\mathcal{P},\mathcal{Q}}^{-1} \right\|_\infty \left\| \begin{bmatrix} (\hat{\mathcal{B}}_i^\circ)^T \\ 1_m^T \end{bmatrix}_{\mathcal{P},\mathcal{Q}} - \begin{bmatrix} (\mathcal{B}_i^\circ)^T \\ 1_m^T \end{bmatrix}_{\mathcal{P},\mathcal{Q}} \right\|_\infty} \\
&\leq \frac{Sen((\mathcal{B}_i^\circ)^T)}{1 - Sen((\mathcal{B}_i^\circ)^T)\|d_i(\mathcal{B},\hat{\mathcal{B}})\|_\infty}.
\end{aligned} \tag{78}
$$

We have that

$$Sen((\hat{\mathcal{B}}_i^\circ)^T) \leq \frac{Sen((\mathcal{B}_i^\circ)^T)}{1 - dSen((\mathcal{B}_i^\circ)^T)}, \tag{79}$$

which leads to the final result:

$$
\begin{aligned}
&V_i^+(A,\hat{\mathcal{B}}) - V_i^-(A,\hat{\mathcal{B}}) \\
&\leq 2d\|A_{:,i}\|_\infty Sen((\hat{\mathcal{B}}_i^\circ)^T) \\
&\leq \frac{2d\|A_{:,i}\|_\infty Sen((\mathcal{B}_i^\circ)^T)}{1 - dSen((\mathcal{B}_i^\circ)^T)} \\
&\leq 4d\|A_{:,i}\|_\infty Sen((\mathcal{B}_i^\circ)^T),
\end{aligned} \tag{80}
$$

where the last inequality holds because $1 - dSen((\mathcal{B}_i^\circ)^T) \geq \frac{1}{2}$. □

## C.5. Refined Statement and Proof of Theorem 5.17

We first expand the big $O(\cdot)$ notation in the statement of Theorem 5.17 and obtain the following theorem:

**Theorem C.5.** *Under Assumption 5.12, if $P_1$ has an estimator $\hat{\mathcal{B}}$ of $\mathcal{B}$ such that $\|d_i(\mathcal{B},\hat{\mathcal{B}})\|_\infty \leq \epsilon = O(g(T)/T), \forall i$ for some $g(T) = o(T)$, then if $P_2$ is using a $f$-no-regret algorithm $\mathcal{A}_2$ with constant $C$, there exists an algorithm $\mathcal{A}_1$ satisfying:*

$$
\begin{aligned}
StackReg_1(\mathcal{A}_1,\mathcal{A}_2) &= \left( 4\left(\epsilon T + \sqrt{Tf(T)}\right) Sen\left((\mathcal{B}_{i*}^\circ)^T\right) + \frac{C\sqrt{Tf(T)}}{\max_{j,k}\|B(e_j - e_k)\|_\infty} \right) \|A\|_{\max} \\
&= O\left(\sqrt{Tf(T)} + g(T)\right).
\end{aligned} \tag{81}
$$

*Proof.* Consider the algorithm $\mathcal{A}_1$ that commits to the solution $(x^-, i^-)$ to:

$$
\begin{aligned}
\text{maximize}_{i,x} \quad & V_i^-(A, \hat{\mathcal{B}}) = x^T A_{:,i} \\
\text{subject to} \quad & (\hat{\mathcal{B}}_i^\circ)^T x \le -(\epsilon + \tilde{f}(T)/T) 1_{n-1}, \\
& x \in \Delta_m, i \in [n].
\end{aligned}
\tag{82}
$$

where $\tilde{f}(T)$ is some function satisfying $\tilde{f}(T) = o(T)$ and $f(T) = o(\tilde{f}(T))$. Since $\epsilon + \tilde{f}(T)/T \ge \|d_i(\mathcal{B}, \hat{\mathcal{B}})\|_\infty$, Proposition 5.9 guarantees that $e_{i^-}$ is a best response to $x^-$ under $\mathcal{B}$.

Let $(x^*, i^*)$ denote the Stackelberg equilibrium of the game. We have that $E_{i^*}(\mathcal{B}) \ne \emptyset$ and since $\epsilon = O(g(T)/T)$, $\epsilon + \tilde{f}(T)/T$ goes to zero as $T \to \infty$, Assumption 5.12 and Proposition 5.14 guarantees that $V_{i^*}^-(A, \hat{\mathcal{B}})$ is well-defined (and therefore so is $V_{i^*}^+(A, \hat{\mathcal{B}})$) for large enough $T$.

Since $\mathcal{A}_2$ is $f$-no-regret, it holds that

$$
\mathbb{E}_{\mathcal{A}_2} \left[ \sum_{t=1}^T (x^-)^T B(e_{i^-} - y_t) \right] \le C \cdot f(T)
\tag{83}
$$

for some constant $C$. We have that

$$
\begin{aligned}
& \mathbb{E}_{\mathcal{A}_1, \mathcal{A}_2} \left[ \sum_{t=1}^T x_t^T A y_t \right] \\
= & \mathbb{E}_{\mathcal{A}_1, \mathcal{A}_2} \left[ \sum_{t=1}^T (x^-)^T A y_t \right] \\
= & \mathbb{E}_{\mathcal{A}_1, \mathcal{A}_2} \left[ \sum_{t=1}^T (x^-)^T A e_{i^-} \right] + \mathbb{E}_{\mathcal{A}_1, \mathcal{A}_2} \left[ \sum_{t=1}^T (x^-)^T A(y_t - e_{i^-}) \right].
\end{aligned}
\tag{84}
$$

For the first term, notice that for large enough $T$, $\epsilon + \tilde{f}(T)/T \le \frac{1}{2Sen((\mathcal{B}_{i^*}^\circ)^T)}$ and get:

$$
\begin{aligned}
& \mathbb{E}_{\mathcal{A}_1, \mathcal{A}_2} \left[ \sum_{t=1}^T (x^-)^T A e_{i^-} \right] \\
= & T V_{i^-}^-(A, \hat{\mathcal{B}}) \\
\overset{(i)}{\ge} & T V_{i^*}^-(A, \hat{\mathcal{B}}) \\
= & T V_{i^*}^+(A, \hat{\mathcal{B}}) - (T V_{i^*}^+(A, \hat{\mathcal{B}}) - T V_{i^*}^-(A, \hat{\mathcal{B}})) \\
\overset{(ii)}{\ge} & T V(A, \mathcal{B}) - 4(\epsilon T + \tilde{f}(T)) \|A_{:,i^*}\|_\infty Sen((\mathcal{B}_{i^*}^\circ)^T),
\end{aligned}
\tag{85}
$$

where (i) holds since we are taking maximum in (82), and (ii) holds by Proposition 5.9 and Lemma 5.16.

for the second term, since $x^-$ satisfies

$$
(\hat{\mathcal{B}}_{i^-}^\circ)^T x^- \le -(\epsilon + \tilde{f}(T)/T) 1_{n-1},
\tag{86}
$$

we have

$$
\begin{aligned}
(\mathcal{B}_{i^-}^\circ)^T x^- = & (\hat{\mathcal{B}}_{i^-}^\circ)^T x^- + (\mathcal{B}_{i^-}^\circ - \hat{\mathcal{B}}_{i^-}^\circ)^T x^- \\
\overset{(i)}{\le} & (-(\epsilon + \tilde{f}(T)/T) + \|d_{i^-}(\mathcal{B}, \hat{\mathcal{B}})\|_\infty) 1_{n-1} \\
\overset{(ii)}{\le} & -\frac{\tilde{f}(T)}{T} 1_{n-1},
\end{aligned}
\tag{87}
$$

where (i) holds by definition of $d_i(\cdot, \cdot)$ and (51), and (ii) holds by the assumption that $\|d_{i^-}(\mathcal{B}, \hat{\mathcal{B}})\|_\infty \le \epsilon$.

Thus $e_{i-}$ is the unique best response to $x^-$ and Hölder's inequality yields

$$(x^-)^T B \mathbb{E}_{\mathcal{A}_2} \left[ \sum_{t=1}^T (e_{i-} - y_t) \right] \geq \frac{\tilde{f}(T)}{T} \max_{j,k} \| B(e_j - e_k) \|_\infty \left\| \mathbb{E}_{\mathcal{A}_2} \left[ \sum_{t=1}^T (y_t - e_{i-}) \right] \right\|_1. \tag{88}$$

Therefore,

$$\left\| \mathbb{E}_{\mathcal{A}_2} \left[ \sum_{t=1}^T (y_t - e_{i-}) \right] \right\|_1 \leq \frac{C f(T) T}{\tilde{f}(T) \max_{j,k} \| B(e_j - e_k) \|_\infty}. \tag{89}$$

Thus we have:

$$\mathbb{E}_{\mathcal{A}_1, \mathcal{A}_2} \left[ \sum_{t=1}^T (x^-)^T A (y_t - e_{i-}) \right]$$

$$\overset{(i)}{\geq} - \|(x^-)^T A\|_\infty \left\| \mathbb{E}_{\mathcal{A}_2} \left[ \sum_{t=1}^T (y_t - e_{i-}) \right] \right\|_1 \tag{90}$$

$$\overset{(ii)}{\geq} - \|A\|_{\max} \frac{C f(T) T}{\tilde{f}(T) \max_{j,k} \| B(e_j - e_k) \|_\infty},$$

where we apply Hölder again in (i) and use the fact that $x^- \in \Delta_m$ in (ii).

Combining (84), (85) and (90) we obtain as $T \to \infty$:

$$\mathbb{E}_{\mathcal{A}_1, \mathcal{A}_2} \left[ \sum_{t=1}^T x_t^T A y_t \right]$$

$$\geq TV(A, \mathcal{B}) - 4(\epsilon T + \tilde{f}(T)) \|A_{:,i^*}\|_\infty Sen((\mathcal{B}_{i^*}^\circ)^T)$$

$$- \|A\|_{\max} \frac{C f(T) T}{\tilde{f}(T) \max_{j,k} \| B(e_j - e_k) \|_\infty} \tag{91}$$

$$\geq TV(A, \mathcal{B}) - 4(g(T) + \tilde{f}(T)) \|A_{:,i^*}\|_\infty Sen((\mathcal{B}_{i^*}^\circ)^T)$$

$$- \|A\|_{\max} \frac{C f(T) T}{\tilde{f}(T) \max_{j,k} \| B(e_j - e_k) \|_\infty}.$$

taking $\tilde{f}(T) = \sqrt{T f(T)}$ we have:

$$StackReg_1(\mathcal{A}_1, \mathcal{A}_2)$$

$$\leq 4 \left( \epsilon T + \sqrt{T f(T)} \right) \|A_{:,i^*}\|_\infty Sen\left( (\mathcal{B}_{i^*}^\circ)^T \right) + \|A\|_{\max} \frac{C \sqrt{T f(T)}}{\max_{j,k} \| B(e_j - e_k) \|_\infty}$$

$$\leq \left( 4 \left( \epsilon T + \sqrt{T f(T)} \right) Sen\left( (\mathcal{B}_{i^*}^\circ)^T \right) + \frac{C \sqrt{T f(T)}}{\max_{j,k} \| B(e_j - e_k) \|_\infty} \right) \|A\|_{\max} \tag{92}$$

$$= O(g(T) + \sqrt{T f(T)}).$$

This completes the proof of Theorem 5.17. $\qquad \square$

## D. Lower Bound on Stackelberg Regret against General $f$-no-regret Learners

In this section we provide Theorem D.1, which shows that even if the optimizer knows $B$, the learner still has a no-regret algorithm with regret budget $f$ that can lead to a $\sqrt{Tf(T)}$ Stackelberg regret of $P_1$, indicating that our Stackelberg regret bound in Theorem 5.17 is essentially optimal.

**Theorem D.1.** *Consider a given function $f(T) = o(T)$ which serves as the regret budget of $P_2$, there exists a game instance $G = (A, B)$ that satisfies Assumption 5.12 and an $f$-no-regret learner algorithm $\mathcal{A}_2$ that can be used by $P_2$ such that for all mixed strategy $x \in \Delta_m$, the non-adaptive algorithm $\mathcal{A}_1$ that plays $x_t = x$ at all time steps has Stackelberg regret at least $\Omega(\sqrt{Tf(T)})$.*

*Proof.* Consider the game instance $G = (A, B)$ where:

$$A = \begin{bmatrix} 0 & 0 \\ 3 & 1 \end{bmatrix}, B = \begin{bmatrix} 1 & 0 \\ 0 & 1 \end{bmatrix}, \tag{93}$$

whose unique Stackelberg equilibrium is:

$$x^* = (\frac{1}{2}, \frac{1}{2})^T, y^* = (1, 0)^T \tag{94}$$

with a Stackelberg value $V(A, B) = \frac{3}{2}$. For an algorithm $\mathcal{A}_1$ that outputs a fixed optimizer action $x = (x_1, x_2)^T$ and given a sequence $\{y_t\}_{t=1}^T$ of learner actions, the learner regret can be expressed as:

$$Reg_2(\{x_t = x, y_t\}_{t=1}^T) = T \max\{x_1, x_2\} - \sum_{t=1}^T x^T y_t. \tag{95}$$

Consider the learner algorithm $\mathcal{A}_2$ to be:

1. If $x_2 \geq x_1$, play $(0, 1)^T$;

2. Otherwise, play $(0, 1)^T$ for $\frac{f(T)}{1 - 2x_2}$ rounds, and $(1, 0)^T$ for the remaining $T - \frac{f(T)}{1 - 2x_2}$ rounds.

The learner regret of $\mathcal{A}_2$ if $x_2 \geq x_1$ is:

$$Reg_2(\mathcal{A}_2, \{x_t = x\}_{t=1}^T) = Tx_2 - Tx_2 = 0, \tag{96}$$

and if $x_2 < x_1$, we have:

$$\begin{aligned} Reg_2(\mathcal{A}_2, \{x_t = x\}_{t=1}^T) =& Tx_1 - \frac{x_2 f(T)}{1 - 2x_2} - (T - \frac{f(T)}{1 - 2x_2})x_1 \\ =& T(1 - x_2) - \frac{x_2 f(T)}{1 - 2x_2} - (T - \frac{f(T)}{1 - 2x_2})(1 - x_2) \\ =& \frac{(1 - x_2)f(T)}{1 - 2x_2} - \frac{x_2 f(T)}{1 - 2x_2} \\ =& f(T), \end{aligned} \tag{97}$$

and therefore $\mathcal{A}_2$ is $f$-no-regret.

If the optimizer wants to achieve sublinear Stackelberg regret, $x$ must satisfy $x_2 < x_1$, or otherwise $\mathcal{A}_2$ will stick to $(0, 1)^T$ and incur a $\Theta(T)$ Stackelberg regret, now we calculate the Stackelberg regret of the optimizer when $x_2 < x_1$:

$$\begin{aligned} StackReg_1(\{x_t = x, y_t\}_{t=1}^T) =& \frac{3}{2}T - (3(T - \frac{f(T)}{1 - 2x_2}) + \frac{f(T)}{1 - 2x_2}) \cdot x_2 \\ =& (\frac{3}{2} - 3x_2)T + \frac{2x_2}{1 - 2x_2}f(T). \end{aligned} \tag{98}$$

The minimum Stackelberg regret over $x_2 \in [0, \frac{1}{2})$ can be achieved by taking

$$x_2 = \frac{1}{2} - \sqrt{\frac{f(T)}{6T}}, \tag{99}$$

where $StackReg_1(\{x_t = x, y_t\}_{t=1}^T) = \Theta(\sqrt{Tf(T)})$, so in general the Stackelberg regret is $\Omega(\sqrt{Tf(T)})$. $\qquad\square$

# E. Algorithms and Proofs for Section 6

Before we start presenting the algorithms and proofs, we first prove an auxiliary lemma which suggests that in an explore-then-commit style algorithm, any interaction history that has length $o(T)$ before committing will have no impact on the asymptotic learner regret, stated formally as follows:

**Lemma E.1.** *Consider an optimizer action $\tilde{x} \in \Delta_m$, if the optimizer action sequence $\{x_t\}_{t=1}^T$ satisfies $x_t = \tilde{x}, \forall t > \tau$ for some $\tau = O(f(T))$, then the interaction sequence $\{x_t, y_t\}_{t=1}^T$ is $f$-no-regret if and only if*

$$Reg_2(\{x_t, y_t\}_{t=\tau+1}^T) \leq C \cdot f(T) \tag{100}$$

*for some constant $C$, and consequently, $\mathcal{A}_2$ is $f$-no-regret on $\{x_t\}_{t=1}^T$ if and only if*

$$\tilde{x}^T B \mathbb{E}_{\mathcal{A}_2} \left[ \sum_{t=\tau+1}^T (\tilde{y} - y_t) \right] \leq C \cdot f(T) \tag{101}$$

*for some constant $C$.*

*Proof.* Let $\tilde{y} \in BR(B, \tilde{x})$ be a best response to $\tilde{x}$. For the "if" direction, on observing that:

$$
\begin{aligned}
Reg_2(\{x_t, y_t\}_{t=1}^T) &= \max_{y \in \Delta_n} \sum_{t=1}^T x_t^T B y - \sum_{t=1}^T x_t^T B y_t \\
&\leq \max_{y \in \Delta_n} \sum_{t=1}^\tau x_t^T B y - \sum_{t=1}^\tau x_t^T B y_t + \max_{y \in \Delta_n} \sum_{t=\tau+1}^T x_t^T B y - \sum_{t=\tau+1}^T x_t^T B y_t \\
&= Reg_2(\{x_t, y_t\}_{t=1}^\tau) + Reg_2(\{x_t, y_t\}_{t=\tau+1}^T) \\
&\leq \tau \|B\|_{\max} + C \cdot f(T) \\
&\leq C' \cdot f(T)
\end{aligned}
\tag{102}
$$

for some constant $C'$ where we have used the fact that $\tau = O(f(T))$, and therefore $\{x_t, y_t\}_{t=1}^T$ is $f$-no-regret. For the "only if" direction, notice that

$$
\begin{aligned}
Reg_2(\{x_t, y_t\}_{t=1}^T) &= \max_{y \in \Delta_n} \sum_{t=1}^T x_t^T B y - \sum_{t=1}^T x_t^T B y_t \\
&\geq \sum_{t=1}^T x_t^T B (\tilde{y} - y_t) \\
&= \sum_{t=1}^\tau x_t^T B (\tilde{y} - y_t) + \tilde{x}^T B \sum_{t=\tau+1}^T (\tilde{y} - y_t) \\
&\geq -\tau \|B\|_{\max} + Reg_2(\{x_t, y_t\}_{t=\tau+1}^T).
\end{aligned}
\tag{103}
$$

If we want $Reg_2(\{x_t, y_t\}_{t=1}^T) \leq C \cdot f(T)$ for some constant $C$, we must have:

$$Reg_2(\{x_t, y_t\}_{t=\tau+1}^T) \leq C \cdot f(T) + \tau \|B\|_{\max} \leq C' f(T) \tag{104}$$

for some constant $C'$, which completes the proof. The proof of (101) follows directly by taking expectation over all $\{y_t\}_{t=1}^T$ trajectories generated by $\mathcal{A}_2$. $\square$

## E.1. Algorithm 1, Detailed Version and Proof of Theorem 6.2 and Discussion

We present the pseudocode for Algorithm 1 as follows:

Here $\texttt{test}(\cdot)$ is a procedure that compares the learner's response at two consecutive time steps to determine the underlying best response to some optimizer action $x = (p, 1-p)$, as shown in Algorithm 2, and $\texttt{BinarySearch}(\cdot, \cdot, d)$ is a procedure that approximates the optimizer action $x^*$ at which the learner is indifferent up to an error margin $d$, as shown in Algorithm 3.

We also provide the detailed version of Theorem 6.2 as follows:

---

**Algorithm 1** Playing Against Ascending Learner

---

   **Input:** Accuracy margin $d$.
   Run $BR(0) \leftarrow \texttt{test}(0)$ and $BR(1) \leftarrow \texttt{test}(1)$.
   **if** $\texttt{test}(0) = \texttt{test}(1)$ **then**
      Let $y^* = \texttt{test}(0) = \texttt{test}(1)$.
      Compute $\tilde{x} \leftarrow \arg\max_{x \in \Delta_m} x^T A y^*$.
   **else**
      $p_L \leftarrow 1[BR(1) = (1,0)]$, $p_R \leftarrow 1[BR(1) = (0,1)]$.
      $p_L^*, p_R^* \leftarrow \texttt{BinarySearch}(p_L, p_R, d)$.
      **if** $p_L^* < p_R^*$ **then**
         $E_1^- \leftarrow [0, \max\{p_L^* - d, 0\}]$;
         $E_2^- \leftarrow [\min\{1, p_R^* + d\}, 1]$.
      **else**
         $E_2^- \leftarrow [0, \max\{p_R^* - d, 0\}]$;
         $E_1^- \leftarrow [\min\{1, p_L^* + d\}, 1]$.
      **end if**
      Compute $\tilde{x}$ through (9) and (10).
   **end if**
   Stick to $\tilde{x}$ for all remaining time steps.

---

**Algorithm 2** $\texttt{test}$

---

   **Input:** Action parameter $p \in [0, 1]$
   Use $t$ to denote the current timestep.
   Play $x_t = (p, 1 - p)^T$ and observe $y_t = (q_t, 1 - q_t)^T$.
   Play an arbitrary $x_{t+1}$ and observe $y_{t+1} = (q_{t+1}, 1 - q_{t+1})^T$.
   **if** $q_{t+1} > q_t$ **then**
      **return** $(1, 0)^T$
   **else**
      **return** $(0, 1)^T$
   **end if**

---

**Theorem E.2.** *Suppose $m = n = 2$ and the payoff matrix $B$ does not contain identical columns. For some chosen parameter $d$, if either one facet is empty, or each facet has diameter at least $d$ and $P_2$ uses an ascent algorithm $\mathcal{A}_2$ that is $f$-no-regret, Algorithm 1 with accuracy margin $d$ achieves a Stackelberg regret of at most*

$$\left( 4 + \frac{C}{\epsilon_1} f(T) \right) \|A\|_{\max} \tag{105}$$

*if the learner has a strictly dominated action, where $\epsilon_1 = \min_{x \in \Delta_m} x^T B(e_1 - e_2)$, and*

$$\left( -2 \log d + 6 + 2Td + \frac{2Cf(T)}{\epsilon_2 d} \right) \|A\|_{\max} \tag{106}$$

*otherwise, where $\epsilon_2$ is a constant that depends only on $B$. Under either case, $C$ is a constant that depends only on the learner regret constant and payoff matrix. This indicates that the Stackelberg regret is at most $O(\frac{f(T)}{d} + dT - \log d)$ as long as $d = \Omega(\exp(-f(T)))$.*

*Proof.* If one of the facet is empty, we would have $\texttt{test}(0) = \texttt{test}(1)$. In this case, despite the first $4$ interaction steps used by $2$ $\texttt{test}$ calls, Algorithm 1 will output the Stackelberg equilibrium strategy for all subsequent actions.

Now we analyze the Stackelberg regret against an $f$-no-regret algorithm $\mathcal{A}_2$. W.l.o.g suppose $e_2$ is the strictly dominated action, since $e_2$ is strictly dominated, we would have $x^T B e_1 - x^T B e_2 > 0$ for all $x \in \Delta_m$, there exists a constant $\epsilon$ that depends only on $B$ (but not $d$), such that:

$$\tilde{x}^T B e_1 - \tilde{x}^T B e_2 \geq \epsilon. \tag{107}$$

---

**Algorithm 3** `BinarySearch`

---

**Input:** Interval endpoints $p_L, p_R$, accuracy margin $d$.

**if** $|p_L - p_R| \leq d$ **then**

    **return** $p_L, p_R$

**end if**

$BR \leftarrow \texttt{test}(\frac{p_L + p_R}{2})$.

**if** $BR = (1,0)^T$ **then**

    **return** $\texttt{BinarySearch}(\frac{p_L + p_R}{2}, p_R, d)$

**else**

    **return** $\texttt{BinarySearch}(p_L, \frac{p_L + p_R}{2}, d)$

**end if**

---

Therefore, by Lemma E.1 the sequence should satisfy:

$$Reg_2(\{x_t, y_t\}_{t=5}^T) = \tilde{x}^T B \sum_{t=5}^T (e_1 - y_t) \leq C \cdot f(T) \tag{108}$$

for some constant $C$, therefore, we have:

$$\|\sum_{t=5}^T (e_1 - y_t)\|_\infty \leq \frac{C}{\epsilon} f(T). \tag{109}$$

This would lead to an upper bound on the Stackelberg regret:

$$StackReg_1(\mathcal{A}_1, \mathcal{A}_2) \leq 4V(A, B) - \sum_{t=1}^4 x_t^T A y_t + \tilde{x}^T A \sum_{t=5}^T (e_1 - y_t)$$

$$\leq 4\|A\|_{\max} + \|\tilde{x}^T A\|_1 \|\sum_{t=5}^T (e_1 - y_t)\|_\infty \tag{110}$$

$$\leq 4\|A\|_{\max} + \|A\|_{\max} \frac{C}{\epsilon} f(T).$$

If neither facets are empty, the binary search phase shrinks the interval length from $1$ to $d$, which requires at most $-\log d + 1$ calls of the `BinarySearch` function. Each Binary search calls the procedure `test` for at most one time, which corresponds to at most two interaction time steps. Therefore, the total number of time steps in the binary search phase is at most $-2 \log d + 6$, leading to a Stackelberg regret of at most $(-2 \log d + 6)\|A\|_{\max}$, which is $O(f(T))$.

Since each facet has length at least $d$, the pessimistic facets computed by $E_1^-$ and $E_2^-$ satisfies $d_H(E_i, E_i^-) \leq 2d$ and $\inf_{x \in E_i^-, x' \in E_j} \|x - x'\|_1 \geq d$ for $i = 1, 2$. Therefore, combining Lemma E.1 and Theorem 5.5 we obtain:

$$StackReg_1(\mathcal{A}_1, \mathcal{A}_2) \leq \lfloor -2 \log d + 6 \rfloor V(A, B) - \sum_{t=1}^{\lfloor -2 \log d + 6 \rfloor} x_t^T A y_t + \tilde{x}^T A \sum_{t=\lfloor -2 \log d + 6 \rfloor + 1}^T (e_1 - y_t)$$

$$\leq (-2 \log d + 6)\|A\|_{\max} + \|\tilde{x}^T A\|_1 \|\sum_{t=\lfloor -2 \log d + 6 \rfloor + 1}^T (e_1 - y_t)\|_\infty \tag{111}$$

$$\leq (-2 \log d + 6)\|A\|_{\max} + \left(2Td + \frac{2Cf(T)}{\epsilon d}\right) \|A\|_{\max}.$$

$\square$

**Discussion on matrix reconstruction view.** Observe that by Definition 5.6 multiplication by a positive constant and shifting an all-one vector in a row preserves the equivalence class, there are only three possible forms of payoff matrices $B$:

$$\begin{bmatrix} 0 & \lambda \\ 0 & 1 \end{bmatrix}; \begin{bmatrix} 0 & \lambda \\ 0 & -1 \end{bmatrix}; \begin{bmatrix} 0 & \lambda \\ 0 & 0 \end{bmatrix}. \tag{112}$$

In the third case (and the instances where $\lambda = 0$) Assumption 5.12 is not satisfied, indicating that this $B$ instance is not learnable. So we focus on the first two cases where $\lambda \neq 0$.

For the first case where

$$B = \begin{bmatrix} 0 & \lambda \\ 0 & 1 \end{bmatrix} \tag{113}$$

for $x_t = (p, 1-p)$ we know that if $\lambda > 0$ then $y^* = (0,1)$ is a dominant action, and when $\lambda \leq 0$ we have:

$$\begin{cases} p < \frac{1}{1-\lambda} & (0,1) \text{ is the best response;} \\ p = \frac{1}{1-\lambda} & \text{all actions are equivalent;} \\ p > \frac{1}{1-\lambda} & (1,0) \text{ is the best response.} \end{cases} \tag{114}$$

Similarly for the second case where

$$B = \begin{bmatrix} 0 & \lambda \\ 0 & -1 \end{bmatrix}, \tag{115}$$

if $\lambda < 0$ then $(1,0)$ is a dominant action and when $\lambda \geq 0$ we have:

$$\begin{cases} p < \frac{1}{1+\lambda} & (1,0) \text{ is the best response;} \\ p = \frac{1}{1+\lambda} & \text{all actions are equivalent;} \\ p > \frac{1}{1+\lambda} & (0,1) \text{ is the best response.} \end{cases} \tag{116}$$

Now we have reduced the problem to identifying the matrix type and finding $\lambda$ (or equivalently, $p^* := \frac{1}{1\pm\lambda}$ where all actions are equivalent to the learner).

### E.2. Algorithm 4, Detailed Version and Proof of Theorem 6.3

We present the pseudocode for Algorithm 4 as follows, where the `ExploreRow` procedure shown in Algorithm 5.

---
**Algorithm 4** Playing Against Mirror Descent

**Input:** Per-row exploration step $k$.
**for** $i = 1, 2, \ldots, m$ **do**
$\quad \hat{B}_i \leftarrow \texttt{ExploreRow}(e_i, k)$
**end for**
Construct estimation $\hat{B} = \begin{bmatrix} \hat{B}_1 & \hat{B}_2 & \ldots & \hat{B}_m \end{bmatrix}^T$
Compute the equivalence class $\hat{\mathcal{B}}$ and commit to $\tilde{x}$ through (82) with $\tilde{f}(T) = \sqrt{Tf(T)}$.

---

---
**Algorithm 5** `ExploreRow`

**Input:** Action $x \in \Delta_m$, exploration step $k$.
**for** $\tau = 1, 2, \ldots, k+1$ **do**
$\quad$ Play $x$, observe the follower action $y_\tau$ in this time step.
**end for**
**return** $\hat{B}_i = \frac{1}{k}\sum_{\tau=1}^{k} -\eta_\tau \nabla_y D(y_{\tau+1} \| y_\tau)$

---

The detailed version of Theorem 6.3 is presented as follows:

**Theorem E.3.** *If the learner payoff matrix $B$ statisfies the assumptions needed in Theorem 5.17, $P_2$ follows update rule (25), and each entry $\xi_{t,i}$ is i.i.d. $R$-sub-Gaussian, then with probability at least $1 - \delta$, $P_1$ using Algorithm 4 with $k = (T/g(T))^2 2R^2 \log(2mn/\delta)$, incurs Stackelberg regret of at most*

$$\left( m(1+k) + 4\left( \frac{8ng(T)}{\max_{j,k}\|B(e_j - e_k)\|_\infty} + \sqrt{Tf(T)} \right) Sen((\mathcal{B}_{i^*}^\circ)^T) + \frac{C\sqrt{Tf(T)}}{\max_{j,k}\|B(e_j - e_k)\|_\infty} \right) \|A\|_{\max}. \tag{117}$$

*where $C$ is a constant that depends only on the learner regret constant and payoff matrix.*

*Proof.* The Lagrangian of this problem can be written as:

$$L(y, \lambda, \mu) = \eta_t D(y\|y_t) - (x_t^T B + \xi_t^T)y - \lambda^T y + \mu(\sum_i y_i - 1). \tag{118}$$

Since $y_{t+1}$ is the optimal solution to (25), the KKT condition yields:

$$\eta_t \nabla_{t+1} y D(y_{t+1}\|y_t) - B^T x_t - \lambda + \mu 1 - \xi_t = 0;$$
$$y_{t+1,i} \cdot \lambda_i = 0, \forall i \in [n]. \tag{119}$$

Since the Bregman divergence regularizer satisfies $\nabla_y D(y_{t+1}\|y_t) \to \infty$ if there exists $i \in [n]$ such that $y_{t+1,i} \to 0$, we can deduce that $\lambda = 0$, and the condition becomes:

$$\eta_t \nabla_y D(y_{t+1}\|y_t) - B^T x_t + \mu 1 - \xi_t = 0. \tag{120}$$

Let $h_t$ denote $h_t := \eta_t \nabla_y D(y_{t+1}\|y_t)$, we know that

$$B^T x_t = h_t + \mu_t 1 - \xi_t, \tag{121}$$

here we used $\mu_t$ instead of $\mu$ to indicate the different Lagrange multipliers at different time steps. If we set $x_t = e_i$, we obtain:

$$B_i = h_t + \mu_t 1 - \xi_t, \tag{122}$$

where $B_i^T$ is the $i$-th row of $B$, so if we fix $x_t = e_i$ for $t = 1$ to $k$, we have the following estimation of $B_i$:

$$\hat{B}_i = \frac{1}{k} \sum_{t=1}^{k} h_t \tag{123}$$

with error term:

$$B_i - \hat{B}_i = \frac{1}{k} \sum_{t=1}^{k} \mu_t 1 - \frac{1}{k} \sum_{t=1}^{k} \xi_t. \tag{124}$$

The first term doesn't affect the equivalence class of $B$, and the second term diminishes over time. Hence we can obtain an estimation with arbitrarily small error using uniform exploration. Assuming that each entry $\xi_{t,i}$ is i.i.d. $R$-sub-Gaussian, we obtain through Chernoff bound:

$$\Pr\left(\frac{1}{k}\|\sum_{t=1}^{k} \xi_t\|_\infty \leq \epsilon\right) \leq 2n \exp\left(-\frac{k\epsilon^2}{2R^2}\right). \tag{125}$$

If we take $k = \frac{2R^2}{\epsilon^2} \log \frac{2mn}{\delta}$ it holds with probability at least $1 - \delta/m$ that

$$\|B_i - \hat{B}_i - \frac{1}{k} \sum_{t=1}^{k} \mu_t 1\|_\infty \leq \epsilon. \tag{126}$$

Combining this with Theorem 5.17, we take $\epsilon = \Theta(g(T)/T)$, the number of steps to explore one row of $B$ would be:

$$k = \left(\frac{T}{g(T)}\right)^2 2R^2 \log \frac{2mn}{\delta}. \tag{127}$$

When $g(T) = \Omega(\sqrt{T})$ we would have $k = o(T)$, indicating that the exploration cost would also be sublinear in $T$. The length of the exploration phase consists of at most $m(k + 1)$ steps and achieves an estimation $\hat{B}$ in the equivalence class $\hat{\mathcal{B}}$. We first give an upper bound on $\|d_i(\mathcal{B}, \hat{\mathcal{B}})\|$ for all $i$. Notice that:

$$\max_{j_1,j_2} \|B_{:,j_1} - B_{:,j_2}\|_\infty - \max_{j_1,j_2} \|\hat{B}_{:,j_1} - \hat{B}_{:,j_2}\|_\infty$$
$$\leq \max_{j_1,j_2}\{\|B_{:,j_1} - B_{:,j_2}\|_\infty - \|\hat{B}_{:,j_1} - \hat{B}_{:,j_2}\|_\infty\}$$
$$\leq \max_{j_1,j_2} \|(B - \hat{B})(e_{j_1} - e_{j_2})\|_\infty$$
$$= \max_{j_1,j_2} \max_i |(B_i^T - \hat{B}_i^T)(e_{j_1} - e_{j_2})|$$
$$= \max_{j_1,j_2} \max_i |\bar{\xi}^T(e_{j_1} - e_{j_2})|$$
$$\leq 2\|\bar{\xi}\|_\infty, \tag{128}$$

where $\bar{\xi} = \frac{1}{k}\sum_t \xi_t$ for all $t$ in this `ExploreRow` function call. To make notation simpler, in this proof we mildly overload the notation to let:

$$\mathcal{B}_i = \begin{bmatrix} B_{:,1} - B_{:,i} & B_{:,2} - B_{:,i} & \dots & B_{:,n} - B_{:,i} \end{bmatrix},$$
$$\hat{\mathcal{B}}_i = \begin{bmatrix} \hat{B}_{:,1} - \hat{B}_{:,i} & \hat{B}_{:,2} - \hat{B}_{:,i} & \dots & \hat{B}_{:,n} - \hat{B}_{:,i} \end{bmatrix}.$$

We have:

$$
\begin{aligned}
d_i(\mathcal{B}, \hat{\mathcal{B}}) =& \frac{\mathcal{B}_i}{\max_{j,k} \|B_{:,j} - B_{:,k}\|_\infty} - \frac{\hat{\mathcal{B}}_i}{\max_{j,k} \|\hat{B}_{:,j} - \hat{B}_{:,k}\|_\infty} \\
=& \frac{\max_{j_1,j_2} \|\hat{B}_{:,j_1} - \hat{B}_{:,j_2}\|_\infty \mathcal{B}_i - \max_{j_1,j_2} \|B_{:,j_1} - B_{:,j_2}\|_\infty \hat{\mathcal{B}}_i}{\max_{j_1,j_2} \|B_{:,j_1} - B_{:,j_2}\|_\infty \times \max_{j_1,j_2} \|\hat{B}_{:,j_1} - \hat{B}_{:,j_2}\|_\infty} \\
=& \frac{\mathcal{B}_i (\max_{j_1,j_2} \|\hat{B}_{:,j_1} - \hat{B}_{:,j_2}\|_\infty - \max_{j_1,j_2} \|B_{:,j_1} - B_{:,j_2}\|_\infty)}{\max_{j_1,j_2} \|B_{:,j_1} - B_{:,j_2}\|_\infty \times \max_{j_1,j_2} \|\hat{B}_{:,j_1} - \hat{B}_{:,j_2}\|_\infty} \\
&+ \frac{\max_{j_1,j_2} \|B_{:,j_1} - B_{:,j_2}\|_\infty (\mathcal{B}_i - \hat{\mathcal{B}}_i)}{\max_{j_1,j_2} \|B_{:,j_1} - B_{:,j_2}\|_\infty \times \max_{j_1,j_2} \|\hat{B}_{:,j_1} - \hat{B}_{:,j_2}\|_\infty} \\
\leq& \frac{2\|\bar{\xi}\|_\infty \mathcal{B}_i}{\max_{j_1,j_2} \|B_{:,j_1} - B_{:,j_2}\|_\infty \times \max_{j_1,j_2} \|\hat{B}_{:,j_1} - \hat{B}_{:,j_2}\|_\infty} + \frac{\mathcal{B}_i - \hat{\mathcal{B}}_i}{\max_{j_1,j_2} \|B_{:,j_1} - B_{:,j_2}\|_\infty - 2\|\bar{\xi}\|_\infty}.
\end{aligned}
\tag{129}
$$

Since our exploration round $k = \left(\frac{T}{g(T)}\right)^2 2R^2 \log \frac{2mn}{\delta}$ we obtain through union bound that with probability at least $1 - \delta$, for all `ExploreRow` function calls, $\|\bar{\xi}\|_\infty \leq \frac{g(T)}{T}$ which goes to zero as $T \to \infty$, for large enough $T$ we have:

$$
\begin{aligned}
\|d_i(\mathcal{B}, \hat{\mathcal{B}})\|_\infty \leq& \frac{4\|\bar{\xi}\|_\infty \|\mathcal{B}_i\|_\infty}{(\max_{j_1,j_2} \|B_{:,j_1} - B_{:,j_2}\|_\infty)^2} + \frac{2\|\mathcal{B}_i - \hat{\mathcal{B}}_i\|_\infty}{\max_{j_1,j_2} \|B_{:,j_1} - B_{:,j_2}\|_\infty} \\
\leq& \frac{4n\|\bar{\xi}\|_\infty}{\max_{j_1,j_2} \|B_{:,j_1} - B_{:,j_2}\|_\infty} + \frac{4n\|\bar{\xi}\|_\infty}{\max_{j_1,j_2} \|B_{:,j_1} - B_{:,j_2}\|_\infty} \\
\leq& \frac{8n\|\bar{\xi}\|_\infty}{\max_{j_1,j_2} \|B_{:,j_1} - B_{:,j_2}\|_\infty} \\
=& \frac{8n}{\max_{j_1,j_2} \|B_{:,j_1} - B_{:,j_2}\|_\infty} \frac{g(T)}{T} \\
=& \Theta(\frac{g(T)}{T}).
\end{aligned}
\tag{130}
$$

Therefore, combining Lemma E.1 and Theorem 5.17 we obtain:

$StackReg_1(\mathcal{A}_1, \mathcal{A}_2)$

$$
\begin{aligned}
&\leq m(k+1)V(A,B) + \left(4\left(\frac{8ng(T)}{\max_{j,k} \|B(e_j - e_k)\|_\infty} + \sqrt{Tf(T)}\right) Sen\left((\mathcal{B}_{i^*}^\circ)^T\right) + \frac{C\sqrt{Tf(T)}}{\max_{j,k} \|B(e_j - e_k)\|_\infty}\right) \|A\|_{\max} \\
&\leq m(k+1)\|A\|_{\max} + \left(4\left(\frac{8ng(T)}{\max_{j,k} \|B(e_j - e_k)\|_\infty} + \sqrt{Tf(T)}\right) Sen\left((\mathcal{B}_{i^*}^\circ)^T\right) + \frac{C\sqrt{Tf(T)}}{\max_{j,k} \|B(e_j - e_k)\|_\infty}\right) \|A\|_{\max} \\
&= O\left(\left(\frac{T}{g(T)}\right)^2\right) + O(\sqrt{Tf(T)} + g(T)) \\
&= O\left(\sqrt{Tf(T)} + g(T) + \left(\frac{T}{g(T)}\right)^2\right),
\end{aligned}
\tag{131}
$$

which completes the proof. $\qquad\square$

# F. Numerical Experiments

## F.1. Empirical Simulations for Section 6.1

For all experiments in this section, we assume the learner is using Online Gradient descent (OGD) with step size

$$\eta_t = \frac{\eta_0}{\sqrt{t}} \tag{132}$$

For the purpose of properly displaying the interaction and learning process, we choose different $\eta_0$ for different game instances. For each game instance, we compare the performance and learning dynamics for optimizer algorithm being either OGD or Binary Search explore-then-commit (BS, Algorithm 1). For Binary Search, we set the accuracy margin $d = 0.01$. For each game instance, we plot both the payoff and the strategy (indicated by its 0-th entry) of each player at different time steps. We assume optimizer is the row player and learner is the column player.

**Matching pennies.** We first test repeated matching pennies, where the payoff matrices are given by:

$$A = \begin{bmatrix} 1 & -1 \\ -1 & 1 \end{bmatrix}; B = \begin{bmatrix} -1 & 1 \\ 1 & -1 \end{bmatrix}. \tag{133}$$

Both the unique Nash equilibrium and the Stackelberg equilibria all have

$$x = (\frac{1}{2}, \frac{1}{2})^T. \tag{134}$$

We obtain the curve shown in Figure 3.

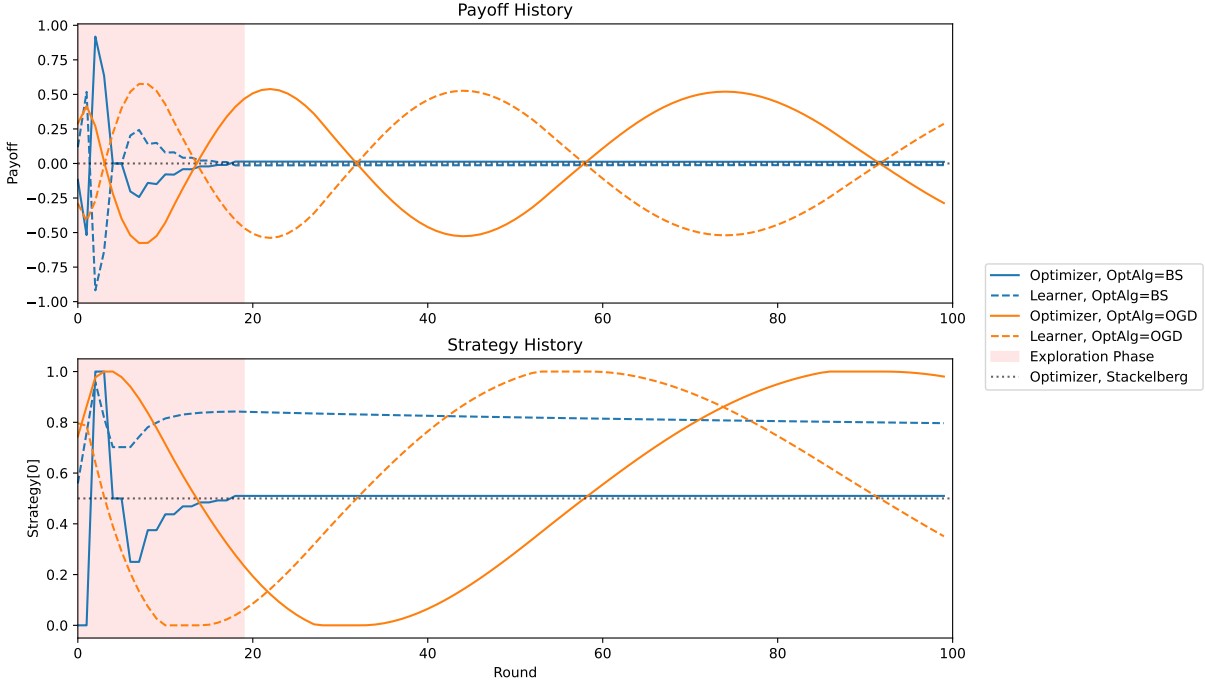

*Figure 3.* Learning dynamics for optimizer algorithms OGD and BS for matching pennies.

Here the blue curves represent the learning dynamics when the optimizer uses binary search (Algorithm 1) with the exploration phase shaded in red. The orange curves represent the dynamics when the optimizer uses OGD. In both optimizer

algorithms, the solid lines are curves for the optimizer and the dashed lines are curves for the learner. We plot the optimizer Stackelberg strategy and payoff in black dotted lines.

We can see that when both players are using OGD, the trajectory keeps oscillating and does not converge to the Nash equilibrium. In comparison, when the optimizer uses BS, it quickly learns its real underlying Stackelberg equilibrium (which is also the Nash) and commits to it, yielding a stable learning dynamics.

**Constructed game instance 1.** Below we show that BS indeed yields a smaller Stackelberg regret than OGD. We construct the following game instance:

$$A = \begin{bmatrix} 5 & 0 \\ 0 & 3 \end{bmatrix}; B = \begin{bmatrix} -2 & 2 \\ 3 & -3 \end{bmatrix}. \tag{135}$$

The unique Stackelberg equilibrium action for the optimizer is:

$$x = (\frac{3}{5}, \frac{2}{5})^T \tag{136}$$

with Stackelberg value 3. We obtain the curve shown in Figure 4. In Figure 4 all curves are drawn with the same line

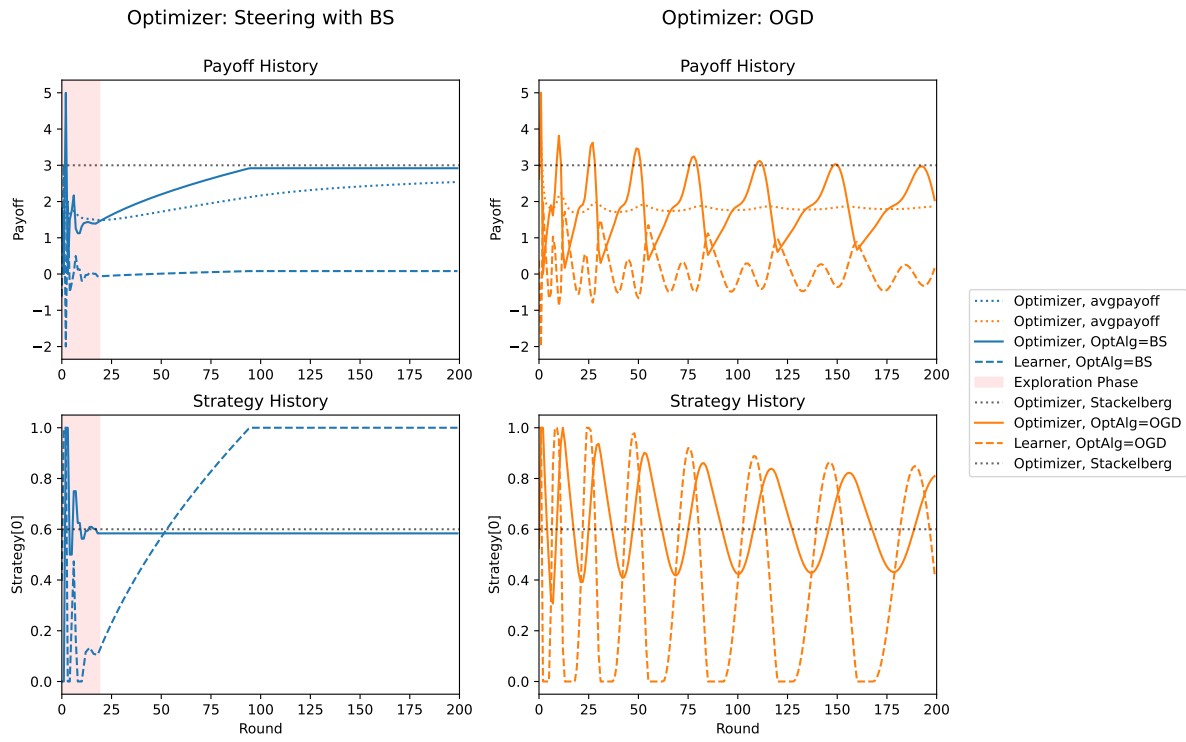

*Figure 4.* Learning dynamics for optimizer algorithms OGD and BS for game instance 1.

style as in Figure 3, in addition we use blue and orange dotted lines to plot the average optimizer payoff for BS and OGD respectively.

We notice again that when the optimizer is using OGD, the algorithm fails to converge. In addition, after the optimizer commits to the pessimistic Stackelberg solution, the learner slowly converges to the best response induced by the Stackelberg equilibrium and steers the optimizer payoff close to the Stackelberg value, which is higher on average than the payoff using SGD.

**Constructed game instance 2.** One may argue that OGD fails because it doesn't converge, however it is not the case. Below we construct a game instance that has a unique Nash equilibrium to which OGD converges, and a unique Stackelberg equilibrium with higher optimizer utility than that of Nash. The game instance is as follows:

$$A = \begin{bmatrix} 2 & 0 \\ 3 & 1 \end{bmatrix}; B = \begin{bmatrix} 1 & 0 \\ 0 & 2 \end{bmatrix}. \tag{137}$$

The unique Nash equilibrium is

$$x = y = (0, 1)^T, \tag{138}$$

while the unique Stackelberg equilibrium is

$$x = (2/3, 1/3)^T, y = (1, 0)^T. \tag{139}$$

The optimizer payoff at Nash is 1, while its Stackelberg value is 2. The simulation result is shown in Figure 5.

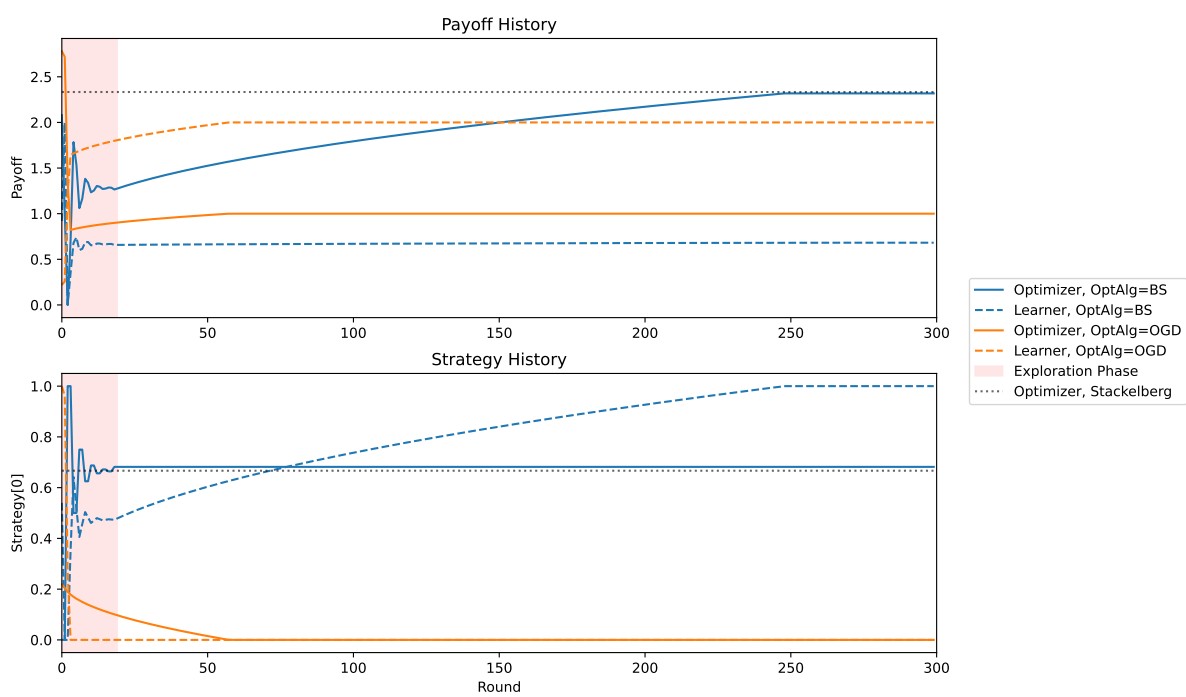

*Figure 5.* Learning dynamics for optimizer algorithms OGD and BS for game instance 2.

The plot above shows that even if both converges, BS and OGD converge to completely different equilibria, while BS always yields a higher payoff and therefore a lower Stackelberg regret.

### F.2. Empirical Simulations for Section 6.2

In this section we show the effectiveness of Algorithm 4. and illustrate the necessity of pessimism. Here we assume the optimizer is using Algorithm 4, but with different pessimism levels $d \in \{0.01, 0.02, 0.05\}$. We assume that the learner is using Stochastic Mirror descent with KL regularizer. For each pure strategy of the optimizer, we set the number of steps for exploration to be $k = 50$. We consider the following game instance:

$$A = \begin{bmatrix} 0 & 1 \\ 5 & 0 \end{bmatrix}; B = \begin{bmatrix} 2 & -2 \\ -3 & 3 \end{bmatrix}. \tag{140}$$

The unique Stackelberg equilibrium of this game is

$$x = (\frac{3}{5}, \frac{2}{5})^T \tag{141}$$

with optimizer payoff 2. We plot the payoffs and strategies of both player at each time step with different $d$ in Figure 6.

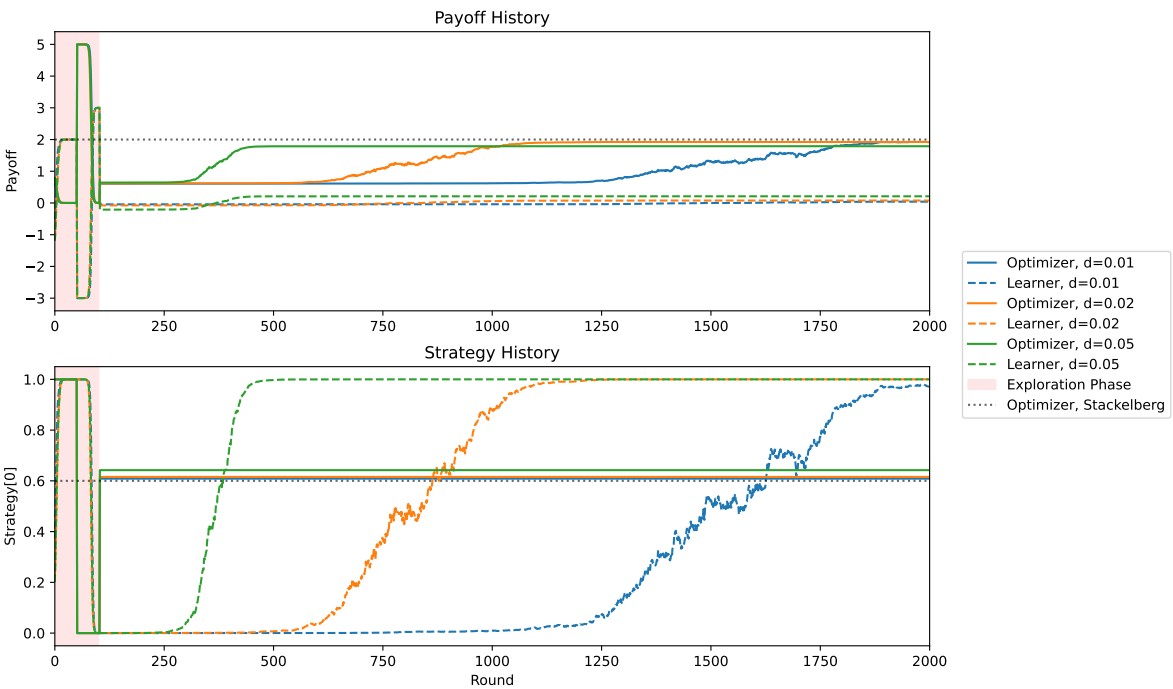

*Figure 6.* Learning dynamics of payoff estimation for different pessimism levels $d$.

In Figure 6, different line colors indicate the learning dynamics of different pessimism levels. The solid lines are optimizer curves while the dashed lines being learner curves. The shaded region indicates the exploration phase and the black dotted line represents the optimizer Stackelberg payoff and strategy.

We can see that for larger $d$, the optimizer is being more pessimistic and chooses an action that is farther away from the Stackelberg equilibrium. This leads to a lower optimizer payoff after the learner converges to the unique best response induced by the action. However, for less pessimistic choices of $d$, since the committed optimizer strategy $\tilde{x}$ is too close to the Stackelberg equilibrium where the learner is indifferent from all mixed strategies, the gradients of the learner payoff with respect $\tilde{x}$ will be extremely small and thus takes a lot longer to converge. Once it hasn't converged the optimizer payoff of committing to $\tilde{x}$ will be smaller, illustrating the effectiveness of being pessimistic.

# G. Proof of Auxiliary Lemmas

## G.1. Proof of Lemma C.2

The dual problem of (71) can be written as:

$$
\begin{aligned}
&\text{minimize} && b^T y \\
&\text{subject to} && A^T y = c, \\
& && y \geq 0,
\end{aligned}
\tag{142}
$$

and that of (72) is:

$$
\begin{aligned}
&\text{minimize} && (b + \delta)^T y \\
&\text{subject to} && A^T y = c, \\
& && y \geq 0.
\end{aligned}
\tag{143}
$$

We can now follow the approach in Section 5.4 and Section 5.5 in (Bertsimas & Tsitsiklis, 1997) to obtain the result: Since both (71) and (72) are feasible and have bounded constraint sets, they have a finite optimal value and a corresponding optimal solution, so do their dual problems. Therefore for each dual problem there exists an optimal solution that is a basic feasible solution (which are the same for both dual problems), denoted by $y_1, y_2, \ldots, y_I$. That is,

$$
V = \min_{i=1,2,\ldots,I} b^T y_i
\tag{144}
$$

and

$$
V(\delta) = \min_{i=1,2,\ldots,I} (b + \delta)^T y_i.
\tag{145}
$$

Therefore we have:

$$
\begin{aligned}
V(\delta) - V &= \min_{i=1,2,\ldots,I} (b + \delta)^T y_i - \min_{i=1,2,\ldots,I} b^T y_i \\
&\leq \max_{i=1,2,\ldots,I} \delta^T y_i.
\end{aligned}
\tag{146}
$$

Since for each $i = 1, 2, \ldots, I$, a linearly independent set of columns $\mathcal{I}$ of $A^T$ of size $n$ captures the basic feasible solution (which is an independent set of rows of $A$) and $y_i$ is specified by $y_i = (A_{\mathcal{I}})^T c$ in its entries within $\mathcal{I}$ and 0 elsewhere, we can rewrite $\delta^T y_i$ as

$$
\delta^T y_i = y_i^T \delta = c^T A_{\mathcal{I}} \delta_{\mathcal{I}},
\tag{147}
$$

which completes the proof.

## G.2. Proof of Lemma C.3

Consider an index set $\mathcal{I}$ containing linearly independent rows of $M$ that satisfies $|\mathcal{I}| = m$.

Since all $m \times m$ submatrices contain $k$ rows in $\begin{bmatrix} (\hat{\mathcal{B}}_i^\circ)^T \\ 1_m^T \\ -1_m^T \end{bmatrix}$ and $m - k$ rows in $-I_m$ for some $0 < k \leq m$. Thus $M_{\mathcal{I}}$ has the form:

$$
M_{\mathcal{I}} = \begin{bmatrix} \tilde{M}_{\mathcal{I}} \\ -e_{\mathcal{I}_{k+1}-n-1}^T \\ -e_{\mathcal{I}_{k+2}-n-1}^T \\ \ldots \\ -e_{\mathcal{I}_m - n - 1}^T \end{bmatrix},
\tag{148}
$$

where $\mathcal{I}_j$ denotes the $j$-th index in $\mathcal{I}$ and $\tilde{M}_{\mathcal{I}}$ contains the first $k$ rows selected from $\begin{bmatrix} (\hat{\mathcal{B}}_i^\circ)^T \\ 1_m^T \\ -1_m^T \end{bmatrix}$. Now we show some properties of $M_{\mathcal{I}}^{-1}$.

First, since $\forall i \in \{k+1, k+2, \ldots, m\}$ and $\forall j \in [m]$, it holds that

$$
[i = j] = I_{ij} \overset{(i)}{=} (M_{\mathcal{I}})_i^T (M_{\mathcal{I}}^{-1})_{:,j} \overset{(ii)}{=} -e_{\mathcal{I}_i - n - 1}^T (M_{\mathcal{I}}^{-1})_{:,j} = -(M_{\mathcal{I}}^{-1})_{\mathcal{I}_i - n - 1, j},
\tag{149}
$$

where (i) comes from the definition of $M_{\mathcal{I}}^{-1}$ and (ii) holds due to (148), we have that:

$$(M_{\mathcal{I}}^{-1})_{\mathcal{I}_i - n - 1} = -e_i^T. \tag{150}$$

Second, consider the product for arbitrary $\Delta \in \mathbb{R}^k$,

$$M_{\mathcal{I}}^{-1} \begin{bmatrix} \Delta \\ 0_{m-k} \end{bmatrix} = \sum_{j=1}^k (M_{\mathcal{I}}^{-1})_{:,j} \Delta_j, \tag{151}$$

whose $i$-th row can be written as:

$$\left( M_{\mathcal{I}}^{-1} \begin{bmatrix} \Delta \\ 0_{m-k} \end{bmatrix} \right)_i = \sum_{j=1}^k (M_{\mathcal{I}}^{-1})_{ij} \Delta_j. \tag{152}$$

Let $\mathcal{E} = \{\mathcal{I}_{k+1} - n - 1, \mathcal{I}_{k+2} - n - 1, \ldots, \mathcal{I}_m - n - 1\}$, we can see from (150) that if $i \in \mathcal{E}$,

$$\left( M_{\mathcal{I}}^{-1} \begin{bmatrix} \Delta \\ 0_{m-k} \end{bmatrix} \right)_i = 0. \tag{153}$$

Also, consider the product of $(M_{\mathcal{I}}^{-1})_{i,:}(M_{\mathcal{I}})_{:,j}$ for $i, j \in [m] \backslash \mathcal{E}$, we have:

$$\begin{aligned} I_{ij} &= (M_{\mathcal{I}}^{-1})_{i,:}^T (M_{\mathcal{I}})_{:,j} \\ &= \sum_{l=1}^k (M_{\mathcal{I}}^{-1})_{i,l} (M_{\mathcal{I}})_{l,j} + \underbrace{\sum_{l=k+1}^m (M_{\mathcal{I}}^{-1})_{i,l} (M_{\mathcal{I}})_{l,j}}_{=0} \\ &= \sum_{l=1}^k (M_{\mathcal{I}}^{-1})_{i,l} (M_{\mathcal{I}})_{l,j}, \end{aligned} \tag{154}$$

where $\sum_{l=k+1}^m (M_{\mathcal{I}}^{-1})_{i,l} (M_{\mathcal{I}})_{l,j} = 0$ because

$$(M_{\mathcal{I}})_{l,j} = 0, \forall l \in \{k+1, \ldots, m\}, j \in [m] \backslash \mathcal{E} \tag{155}$$

as shown in (148). Similarly, for the product of $(M_{\mathcal{I}})_{i,:}(M_{\mathcal{I}}^{-1})_{:,j}$ for $i, j \in [k]$, we have:

$$\begin{aligned} I_{ij} &= (M_{\mathcal{I}})_{i,:}^T (M_{\mathcal{I}}^{-1})_{:,j} \\ &= \sum_{l \in [m] \backslash \mathcal{E}} (M_{\mathcal{I}})_{i,l} (M_{\mathcal{I}}^{-1})_{l,j} + \underbrace{\sum_{l \in \mathcal{E}} (M_{\mathcal{I}})_{i,l} (M_{\mathcal{I}}^{-1})_{l,j}}_{=0} \\ &= \sum_{l \in [m] \backslash \mathcal{E}} (M_{\mathcal{I}})_{i,l} (M_{\mathcal{I}}^{-1})_{l,j}, \end{aligned} \tag{156}$$

where $\sum_{l \in \mathcal{E}} (M_{\mathcal{I}})_{i,l} (M_{\mathcal{I}}^{-1})_{l,j} = 0$ due to

$$(M_{\mathcal{I}}^{-1})_{l,j} = 0, \forall l \in \mathcal{E}, j \in [k] \tag{157}$$

as indicated by (150).

The two equations (154) and (156) above show that if we consider the submatrix of $(M_{\mathcal{I}})_{[k],[m] \backslash \mathcal{E}}$ consisting of its first $k$ rows and columns in $[m] \backslash \mathcal{E}$, its inverse is equal to the corresponding entries of the inverse of $M_{\mathcal{I}}$. More specifically,

$$(M_{\mathcal{I}}^{-1})_{[m] \backslash \mathcal{E}, [k]} = (M_{\mathcal{I}})_{[k],[m] \backslash \mathcal{E}}^{-1}. \tag{158}$$

Now that we have:

$$\left\| M_{\mathcal{I}}^{-1} \begin{bmatrix} \Delta \\ 0_{m-k} \end{bmatrix} \right\|_\infty = \|(M_{\mathcal{I}}^{-1})_{:,[k]} \Delta\|_\infty \overset{(i)}{=} \|(M_{\mathcal{I}}^{-1})_{[m] \backslash \mathcal{E}, [k]} \Delta\|_\infty = \|(M_{\mathcal{I}})_{[k],[m] \backslash \mathcal{E}}^{-1} \Delta\|_\infty, \tag{159}$$

where again (i) holds due to (157). Since the selection of $\mathcal{I}$ is arbitrary, $\mathcal{E}$ can also be constructed arbitrarily, so the upper bound for $\left\| M_{\mathcal{I}}^{-1} \begin{bmatrix} \Delta \\ 0_{m-k} \end{bmatrix} \right\|_\infty$ would be

$$\max_{\mathcal{I},\mathcal{E}} \|(M_{\mathcal{I}})^{-1}_{[k],[m]\setminus\mathcal{E}}\Delta\|_\infty = \max_{\mathcal{P},\mathcal{Q}} \left\| \begin{bmatrix} (\hat{\mathcal{B}}_i^\circ)^T \\ 1_m^T \end{bmatrix}^{-1}_{\mathcal{P},\mathcal{Q}} \Delta \right\|_\infty , \tag{160}$$

where the maximization is over all $\mathcal{P}, \mathcal{Q}$ that satisfies:

$$\mathcal{P} \subseteq [n], \mathcal{Q} \subseteq [m], |\mathcal{P}| = |\mathcal{Q}|, \begin{bmatrix} (\hat{\mathcal{B}}_i^\circ)^T \\ 1_m^T \end{bmatrix}_{\mathcal{P},\mathcal{Q}} \text{ invertible.} \tag{161}$$

Also, because the entries corresponding to $\pm 1_m^T$ and are not perturbed, we can multiply those constraints by arbitrary nonzero $\epsilon$ and still get the same result, which completes the proof.

### G.3. Proof of Lemma C.4

Given that $\|B^{-1}\|\|\delta B\| < 1$, we can expand the series $(I + B^{-1}\delta B)^{-1}$ as a convergent Neumann series:

$$(I + B^{-1}\delta B)^{-1} = \sum_{k=0}^{\infty} (-B^{-1}\delta B)^k. \tag{162}$$

So that $(I + B^{-1}\delta B)^{-1}$ is well-defined. As a result,

$$(B + \delta B)^{-1} = (B(I + B^{-1}\delta B))^{-1} = (I + B^{-1}\delta B)^{-1}B^{-1} \tag{163}$$

is also well-defined, and further

$$\begin{aligned}
\|(B + \delta B)^{-1}\| &= \|(I + B^{-1}\delta B)^{-1}B^{-1}\| \\
&\leq \|(I + B^{-1}\delta B)^{-1}\|\|B^{-1}\| \\
&= \|B^{-1}\|\|\sum_{k=0}^{\infty}(-B^{-1}\delta B)^k\| \\
&\leq \|B^{-1}\|\sum_{k=0}^{\infty}\|(-B^{-1}\delta B)^k\| \\
&\leq \|B^{-1}\|\sum_{k=0}^{\infty}\|B^{-1}\|^k\|\delta B\|^k \\
&= \frac{\|B^{-1}\|}{1 - \|B^{-1}\|\|\delta B\|},
\end{aligned} \tag{164}$$

which completes the proof.

