# OpenReview forum: "Learning to Steer Learners in Games"
_ICML.cc/2025/Conference — ICML 2025 poster_

### Official Review · Reviewer_2TDF · 2025-03-10

**Overall Recommendation:** 3

**Summary:**

The work extends the work by Deng et al, 2019, in showing that one of the key elements of steering is the knowledge of $B$ (Section 4). The authors give some results for steering under (pessimistic) approximations of the 'best response regions' (Def. 5.1) (Section 5) and show upper and lower bounds (Section 5.3 and 6) on the Stackelberg regret (Def 3.5). This is done under the assumption that the learner uses a specific kind of no-regret algorithm and the optimizer uses a 'explore then commit' strategy.

**Claims And Evidence:**

All claims are supported by proofs. However, some of them I could not verify.

**Essential References Not Discussed:**

The authors might also consider referring to
'AN INFORMATION-THEORETIC APPROACH TO MINIMAX REGRET IN PARTIAL MONITORING', Lattimor, Szepesvári, 2019
after Definition 5.1. The idea and analytic purpose of $C_a$ in the reference and of E_i are very similar (identical in definition).

**Ethical Review Concerns:**

None.

**Experimental Designs Or Analyses:**

This is theoretical work, so no experiments are provided.

**Methods And Evaluation Criteria:**

This is theoretical work, thus, evaluation with benchmark datasets or similar empirical methods is not relevant.

**Other Comments Or Suggestions:**

small:
- In Appendix E, equation (104) and following, it should be $T$ not $\infty$?

**Other Strengths And Weaknesses:**

The paper is mostly well written. I am leaning towards 'accept', however, a few things should be clarified. Especially, I find the use of asymptotic notation and early switch to this in the proofs a bit dangerous. There are some steps I can not see/verify from the proofs due to early hiding of the constants or since some dependencies are not shown explicitly. (Please, see question to the authors).
I am happy to revise my evaluation if my concerns are addressed.

**Questions For Authors:**

How is $d$ related to $\epsilon$ in the proof of Theorem 6.2? In particular, how is it ensured that $\epsilon$ is a constant, although $\epsilon$ depends (?) on $d$ and $d$ depends on $T$? (this would be needed for the last step in (106) to the best of my understanding)

**Relation To Broader Scientific Literature:**

The work complements the work by Deng et al, 2019 and falls within the brought literature of steering no-regret learners in a game.

**Theoretical Claims:**

I checked some proofs, (Appendix A and E)

---

> ### Author Rebuttal · Authors · 2025-04-01
>
> We thank the reviewer for the careful assessment and constructive critiques, which is essential for us to further improve our work. We give section-wise responses and clarifications to the mentioned issues.
>
> ### Essential References Not Discussed:
>
> > *"The authors might also consider referring to 'AN INFORMATION-THEORETIC APPROACH TO MINIMAX REGRET IN PARTIAL MONITORING', Lattimor, Szepesvári, 2019 after Definition 5.1. The idea and analytic purpose of $C_a$ in the reference and of E_i are very similar (identical in definition)."*
>
> Thank you for pointing out the references. The definition of $C_a$ is indeed conveying the same information of characterizing the region of point against which a specific action is a best response, and their discussions on the intuition behind $C_a$ and the illustrations in the appendix also work for our definition of facets. We will add this work into citation in our revised version.
>
> ### Other Strengths And Weaknesses:
>
> > *"... I find the use of asymptotic notation and early switch to this in the proofs a bit dangerous. There are some steps I can not see/verify from the proofs due to early hiding of the constants or since some dependencies are not shown explicitly. (Please, see question to the authors). I am happy to revise my evaluation if my concerns are addressed."*
>
> Here we present the exact bounds for the following theorems, in Theorems 5.4 and 5.15 the learner regret bound is indicated by $Cf(T)$:
>
> Theorem 5.4: $\epsilon>0$ is a constant that depends on $B$, please see our reply to (Comment 1) in **Other Comments Or Suggestions** section to Reviewer abnn for details.
>
> $$
> \left(Td_1+\frac{2Cf(T)}{\epsilon d_2}\right)\|A\|_{\max};
> $$
>
> Theorem 5.15:
>
> $$
> \left(4(\|d _i(\mathcal{B},\hat{\mathcal{B}})\| _{\infty} T+\sqrt{Tf(T)}) Sen((\mathcal{B} _{i^*}^\circ)^T) +\frac{C\sqrt{Tf(T)}}{\max _{j,k} \|B(e_j-e_k)\| _\infty}\right)\|A\| _{\max};
> $$
>
> For the two theorems below, we assume the adjusted learner regret rate after the exploration phase is indicated by $Cf(T)$, see response to Reviewer abnn and Lemma E.1 for details.
>
> Theorem 6.2: If there exists a strictly dominated action, let $\epsilon_1=\min_{x\in\Delta_m} x^T B(e_1-e_2)$,
>
> $$
> \left(4+\frac{C}{\epsilon_1}f(T)\right) \|A\|_{\max};
> $$
>
> Otherwise, $\epsilon_2>0$ is a constant that depends on $B$ (as a result of Theorem 5.4) and $d$ is a tunable parameter of the algorithm (see **Questions For Authors** section):
>
> $$
> \left(-2\log d+5+Td+\frac{2Cf(T)}{\epsilon_2 d}\right)\|A\|_{\max};
> $$
>
> Theorem 6.3: Here $k=\left(T/g(T)\right)^2 2R^2\log(2mn/\delta)$, $g(T)=o(T)$ is a tunable parameter of the algorithm deciding the length of exploration rounds,
>
> $$
> \left(m\left(1+k\right)+4(g(T)+\sqrt{Tf(T)})Sen((\mathcal{B} _{i^*}^\circ)^T)+\frac{C\sqrt{Tf(T)}}{\max _{j,k} \|B(e_j-e_k)\| _\infty}\right)\|A\| _{\max}.
> $$
>
> We will include a version of our theorems in the detailed format with exact bounds in our revision. For the usage of asymptotic notations, please also refer to our reply to (Comment 2) in **Other Comments Or Suggestions** section to Reviewer abnn for details.
>
>
> ### Other Comments Or Suggestions:
>
> > *"In Appendix E, equation (104) and following, it should be $T$ not $\infty$ ?"*
>
> Thank you for pointing out, this is a typo and we will correct this in the revision.
>
> ### Questions For Authors:
>
> > *"How is $d$ related to $\epsilon$ in the proof of Theorem 6.2? In particular, how is it ensured that $\epsilon$ is a constant, although $\epsilon$ depends on $d$ and $d$ depends on $T$? ..."*
>
> We apologize for causing the confusion. Here $d$ is an accuracy margin that is used as an adjustable input to Algorithm 1 which we didn't mention in the main body. The idea is that we do binary search until we have identified the boundary of the facets by an accuracy level of at most $d$. We then stick to the pessimistic Stackelberg equilibrium for the remaining time steps, paying a total cumulative regret of $dT$. The statement between Lines 1412 and 1414 is indeed confusing, and from which "... the assumption that each facet has length at least $d$ indicates that ..." should be deleted from this sentence. What we are trying to say is that since $e_2$ is strictly dominated, we would have $x^TBe_1-x^TBe_2>0,\forall x\in\Delta_m$, so that such constant $\epsilon$ exists and is only dependent on $B$ instead of $d$.
>
> The reason why we leave $d$ in the big $O(\cdot)$ notation is that we want to leave some freedom for choosing $d$ as a function of $T$ for different $f(T)$. For example if we choose $d=\sqrt{f(T)/T}$ the regret bound would be $O(2\sqrt{Tf(T)}+\frac{1}{2}\log\left(\frac{T}{f(T)}\right))$. Here we omit the result $O(f(T))$ obtained in the case where one facet is empty because $O(\frac{f(T)}{d}+dT-\log d)$ is always $\Omega(f(T))$ as long as $f(T)$ is $O(T)$ no matter what $d$ is chosen.
>
> Thank you for recognizing and pointing this out. We will make the above point clear in the revision.

---

### Official Review · Reviewer_abnn · 2025-03-12

**Overall Recommendation:** 3

**Summary:**

This paper studies repeated two-player Stackelberg games where the follower (learner) uses a no-regret algorithm to choose responding actions and the leader (optimizer) aims to play against the learner.  Unlike most previous works (e.g., [Deng et al, 2019]) that assume that the optimizer knows the learner's utility matrix, this work drops this assumption and aims to study under what conditions the optimizer can achieve the Stackelberg equilibrium value:

(Result 1) First, the authors show a negative result: if the learner's utility matrix is unknown and its algorithm can be any no-regret algorithm, then the optimizer cannot achieve the Stackelberg value.

(Result 2) Second, the authors show that, if the optimizer knows the learner's utility matrix or best-response functions up to some small error, then it can almost achieve the Stackelberg value (using the idea of pessimism to deal with the error).

(Result 3) Finally, the authors show that, if the learner's utility matrix is unknown but its algorithm is either any ascent algorithm or a stochastic mirror ascent with known step sizes and regularizer, then the optimizer can learn the learner's utility matrix (approximately) and then achieve the Stackelberg value.

**Claims And Evidence:**

Yes, the claims are supported by clear and convincing theoretical arguments.

**Essential References Not Discussed:**

(Minor weakness 1) Result (2) says that the optimizer can approximately achieve Stackelberg value if he knows the learner's utility matrix up to some small error.  This idea has been known in the literature, e.g., [Gan et al, 2023](https://arxiv.org/abs/2304.14990) (which is cited) and [Lin & Chen, 2024](https://arxiv.org/abs/2402.09721) (which is not cited).  Nevertheless, I am not too worried about this weakness, because Result (2) is more of a building block for Result (3), instead of the main conceptual contribution.

**Experimental Designs Or Analyses:**

No experiments.  This is fine to me given the theoretical nature of this work.

**Methods And Evaluation Criteria:**

The techniques used to prove the three main results all make sense to me.  On the positive side:

(Strength 1) The technique for Result (1) includes a non-trivial construction of two utility matrices and learner's no-regret algorithms that prevent the optimizer from achieving the Stackelberg value.  This is an interesting technical contribution.

(Strength 2) The technique for Result (3) shows how to make use of the specific properties of popular no-regret algorithms (gradient ascent algorithm and stochastic mirror ascent algorithm), combined with binary search, to estimate the utility matrix of the learner.  This idea is interesting and might inspire some follow-up works.

**Other Comments Or Suggestions:**

(Comment 1) The $O(\cdot)$ notation in Theorem 5.4 hides a quantity that depends on the learner's payoff matrix.  According to the proof of Theorem 5.4 in Appendix B.2, the $O(\frac{f(T)}{d_2})$ term is actually $\frac{2f(T)}{\epsilon d_2}$ where $\epsilon$ is a quantity that depends on the matrix B.  I think this quantity is related to the inducibility gap defined in [Gan et al, 2023], and the assumption that $\epsilon > 0$ is required.  Some clarification regarding this hidden quantity would be appreciated.

(Comment 2) In the definition of equivalent classes of utility matrices (Definition 5.5), you allow scaling both $cB'$  and shifting $+\mu 1_n^T$.  Although scaling by $c$ does not change the best-response set $BR(B, x)$, it does change the learner's regret by the same scaling factor $c$.  This scaling factor will then affect the optimizer's Stackelberg regret -- it is another quantity hidden inside the big $O(\cdot)$ notation in Theorem 5.4.

Regarding both comments, hiding quantities (that are not necessarily constants) in big $O(\cdot)$ notation is not a good practice because it might cause analytical errors.  I would suggest the authors to not hide any important quantities in the big $O(\cdot)$ notation.  Relatedly, in the definition of $f$-no-regret (Definition 3.2), instead of defining $Reg_2 \le C\cdot f(T)$ for some constant $C$, directly define $Reg_2 \le f(T)$ without any constant.  In that case, if a learner algorithm is $f$-no-regret on $B$, then it is $cf$-no-regret on the scaled matrix $cB$, and the optimizer's Stackelberg regret will have a term like  $\frac{cf(T)}{c\epsilon d_2} = \frac{f(T)}{\epsilon d_2}$, I think.

(Typo) Page 5, Line 268: capitalize "while".

(Small issue 1): Page 5, "At a Stackelberg equilibrium the learner must be indifferent between multiple pure strategies". I would say "the learner is usually indifferent between multiple pure strategies" instead.  There are some degenerate cases where the learner needs not be indifferent.

(Small issue 2): Page 6, line 324 - 325: maybe mention "$k\ne i$" and clarify that $B_i^\circ$ has $n-1$ columns, with the $i$-column excluded.

**Other Strengths And Weaknesses:**

(Minor weakness 2) The positive Result (3) for special no-regret learning algorithms seems weak.  For example, the result for any ascent algorithm (Theorem 6.2) is restricted to $n=2$ (number of optimizer's actions), which is not very complete.  The result for stochastic mirror ascent (Theorem 6.3) requires the optimizer to know the regularizer and step size) of the learner, which seems to be a strong assumption.

But I think the major contribution of this work is the negative Result (1), and Result (3) is more of a "proof of concept" that some positive results can be obtained for special no-regret learning algorithms.  So, I am not very worried about this weakness.

**Questions For Authors:**

(Q1) Can you clarify what important constants are hidden in the big $O(\cdot)$ notation in Theorem 5.4, Theorem  5.15, Theorem 6.2, and Theorem 6.3?

**Relation To Broader Scientific Literature:**

(Strength 3) The authors did a great job of discussing the relationship with previous works.

There are previous works about playing against no-regret learners in Stackelberg games (mentioned in the "Steering no-regret learners" paragraph) assuming known utility.  There are also previous works (mentioned in the "Learning in stackelberg games" paragraph) about learning the agent's utility matrix for a myopically best-responding agent.  This work simultaneously fills the gaps of these two strands of literature by considering learning unknown utility of a no-regret learning agent.  This is a significant conceptual contribution to both strands of literature.  More interestingly, this work gives a negative result -- learning the unknown utility of a no-regret learning agent is impossible in general.  This negative result can be a good reference for future works that explore conditions for positive results.

**Theoretical Claims:**

Yes, I checked most of the proofs and didn't find any major issues, except for some minor notational unclarity (which I described in Other Comments and Suggestions) that does not affect my judgment for the correctness.

---

> ### Author Rebuttal · Authors · 2025-04-01
>
> We sincerely appreciate the great amount of time and thought the reviewer has invested in evaluating our manuscript. We are grateful for the precise recognition of our contribution and novelty, and the insightful feedbacks allowing us to refine and improve our work. We address each concern section-wise below:
>
> ### Essential References Not Discussed:
>
> > *"(Minor weakness 1)..."*
>
> Thank you for mentioning the related works and your interpretation. While both mentioned works considered potential suboptimal learner responses, neither considers unknown (or up to a small error) learner utility. We would especially like to highlight the difference between our work and [Gan et al., 2023](https://arxiv.org/abs/2304.14990), which also considered a region of $\delta$-optimal response, where the learner's action is with suboptimality at most $\delta$. Although being in a similar to the definition of pessimistic facet in Definition 5.7, their definition relies on the real underlying payoff matrix $B$ when defining the region, compared to our definition that uses the *estimated* payoff matrix. This is saying, that the boundary of their $\delta$-optimal region would be parallel to the real underlying $0$-optimal response region, making their bound easier and more straightforward compared to that in our work. The mismatch in boundary direction in our work requires us to come up with a novel proof of Lemma 5.14 in Appendix C.4, which links the optimistic/pessimistic value difference to the real underlying learner payoff through Definition 5.13, which in our opinion is both a conceptual and technical contribution of our work. We will cite these works and provide further discussion in the revision.
>
> ### Other Strengths And Weaknesses:
>
> > *"(Minor weakness 2) ..."*
>
> We impose the assumption $n=2$ because only in this case can we apply binary search on facet estimation. For $n>2$ case, our method wouldn't directly apply because there could be infinitely many ascent direction for $n>2$, and thus the difference between $y_{t+1}$ and $y_t$ no longer fits the binary search algorithm. This is the main reason why we neede more structure in the follower's algorithm as in the section on Mirror Descent. For the assumption of knowing the regularizer, please refer to **Questions for Authors** part of our reply to Reviewer r9Wp for details.
>
> ### Other Comments Or Suggestions:
>
> > *"(Comment 1) ..."*
>
> Thank you for raising this point. First, we would like to specify that $\epsilon>0$ is encoded within the condition $\inf_{x\in E_{i}^-,x'\in E_j} \Vert x-x'\Vert_1 \geq d_2$, which essnetially suggests that each point in $E_i^-$ is at least $d_2$ away from other facets. More specifically, this means in line 795, $x^-$ is not in $E_j$ for all $j\neq i^-$, and by definition of $E_j$, $\exists j'$, s.t. $(x^-)^TBe_j<(x^-)^TBe_{j'}$. Since such existence of $j'$ holds for all $j\neq i^-$, we can deduce that $(x^-)^TB e_{i^-}-(x^-)^T B e_j>0, \forall j\neq i^-$. For those $j$ where $E_j=\emptyset$, since here $d_2$ satisfies $d_2\leq 2$, we can take $\epsilon_1=\frac{1}{2}\min_{x\in\Delta_m,j\neq i^-}\{x^TB e_{i^-}-x^T B e_j\}$ such that equation (44) holds, which is indeed similar to the inducibility gap defined in Definition 5.9 but we are only taking minimum over those $j$ with empty $E_j$, automatically satisfying $\epsilon_1>0$. For other $j$ since $x^-$ is at least $d_2$ away from $E_j$, there must be a constant $\epsilon_2> 0$ such that $(x^-)^TB e_{i^-}-(x^-)^T B e_j\geq \epsilon_2 d_2$. We proceed the proof by taking $\epsilon=\min \{\epsilon_1,\epsilon_2\}$.
>
> > *"(Comment 2) ..."*
> > *"Regarding both comments..."*
>
> The purpose of allowing a constant scale is that we want to:
>
> 1). Focus on the asymptotic behaviour of the learner's algorithm;
>
> 2). Highlight the *rate* at which the regret is growing in the bound;
>
> 3). Keep the $f$-no-regret property invariant when dealing with the same interaction sequence in the same equivalence class.
>
> Our purpose 2) requires the definition of $f$-no-regret itself does not contain any asymptotic notation, and our purpose 1) requires  any optimizer's exploration phase that is of length $O(f(T))$ won't affect the overall $f$-no-regret property of the learner (see Lemma E.1), which is essential to our proofs of Theorems 6.2 and 6.3. Even if we modify the definition to $Reg_2\leq f(T)$ without the constant, the exploration phase would still induce another constant $C'>C$ which will appear in the final bounds in Section 6 since we have to use Lemma E.1.
>
> > *"(Typo)(Small issue 1)(Small issue 2)"*
>
> Thanks for pointing out the typo and these issues. We will fix them in the revision.
>
> ### Questions For Authors
>
> > *"(Q1) Can you clarify what important constants are hidden in the big $O(\cdot)$ notation in Theorem 5.4, Theorem 5.15, Theorem 6.2, and Theorem 6.3?"*
>
> Please refer to our reply in **Other Strengths And Weaknesses** section to reviewer 2TDF for a more detailed bound without the big $O(\cdot)$ notation.

---

> > ### Comment · Reviewer_abnn · 2025-04-06
> >
> > Thank the authors for the response!  As you responded to me and reviewer 2TDF, the "constants" hidden in the big O notations are indeed non-trivial quantities, and I believe they should be included in the formal results.  I still keep the relatively positive score of 3.

---

> > > ### Author Response · Authors · 2025-04-08
> > >
> > > Dear reviewer,
> > >
> > > We would like to thank you again for your review and suggestions. We will further improve our paper and include formal statements of our results, and we are still willing to engage in possible further discussions and address any potential concerns.
> > >
> > > Best regards,
> > >
> > > Authors

---

### Official Review · Reviewer_DWLF · 2025-03-14

**Overall Recommendation:** 3

**Summary:**

This paper studies the problem of learning the Stackelberg equilibrium against an unknown follower who plays against the leader with some no-regret learning algorithm. They first show a negative result, which shows that this is impossible if no information about the follower is known by the leader. Then they propose sufficient conditions under which the follower can be exploited and design an algorithm that achieves tight sublinear Stackelberg regret. Finally, they provide two specific examples about the update rule of follower’s algorithm which allows the leader to steer to the Stackelberg equilibrium.

**Claims And Evidence:**

yes.

**Essential References Not Discussed:**

I did not notice any essential reference not being discussed

**Experimental Designs Or Analyses:**

n/a

**Methods And Evaluation Criteria:**

yes.

**Other Comments Or Suggestions:**

n/a

**Other Strengths And Weaknesses:**

On the positive side, this paper considers a learning approach to learning the Stackelberg equilibrium against no-regret followers with unknown type information. I found the problem presented by the paper interesting, and relevant. In general, this paper proposed an innovative research topic.

On the other hand, I find the contribution of Section 6 unclear, where learning to steer to the Stackelberg equilibrium with non-sublinear regret. This suggests that the regret does not vanish asymptotically, which raises concerns about the effectiveness of the proposed learning approach. One could apply an efficient learning algorithm from the literature to infer the follower’s type and then leverage the strategy outlined by Deng et al. (2019) to address the problem of strategizing against no-regret learners when the follower’s type is known. This alternative approach would likely achieve similar performance and theoretical properties, making it unclear what additional value the proposed method in Section 6 provides.

**Questions For Authors:**

Can you elaborate on the contribution of non-sublinear Stackelberg regret in Section 6?

**Relation To Broader Scientific Literature:**

This paper is related to the literature of learning in Stackelberg games and strategizing against no-regret learners.

**Theoretical Claims:**

I did not check the correctness of proofs.

---

> ### Author Rebuttal · Authors · 2025-04-01
>
> We would first like to thank the reviewer for thoughtful review and recognition of the novelty in the research topic we proposed. Please find our response listed section-wise below:
>
> ### Other Strengths And Weaknesses:
>
> > *"...I find the contribution of Section 6 unclear, where learning to steer to the Stackelberg equilibrium with non-sublinear regret... "*
> > *"... One could apply an efficient learning algorithm from the literature to infer the follower’s type and then leverage the strategy outlined by Deng et al. (2019) to address the problem... This alternative approach would likely achieve similar performance and theoretical properties..."*
>
>
> There may be some confusion in how we presented our results and we are happy to clarify/simplify the presentation in the revision. We show in Section 6 that the Stackelberg regret is vanishing in both cases. In Theorem 6.2, $d$ is a tunable parameter in our algorithm, indicating the binary search accuracy level. Choosing $d=\sqrt{f(T)/T}$ would yeild a $\sqrt{Tf(T)}$ regret overall. In Theorem 6.3, $g(T)=o(T)$ and thus the regret vanishes asymptotically there as well. We tried to keep our results general but can clarify/instantiate these quantities in the revision to highlight our results.
>
> The goal of section 6 was to give sufficient conditions under which steering a learning agent to a Stackelberg equilibrium is possible (in the sense that the Stackelberg regret is vanishing in time). Our impossibility results hints that this is a hard goal against arbitrary no-regret learers, and so we wanted to understand what further restrictions we could make on the follower for the leader to succeed. To that end, we identified two sub-classes of no-regret algorithms in which our problem can be efficiently addressed: (1). ascending learners  in games with two actions, and (2). general mirror descent learners (with known regularizers) in general games. While the class in (1). is quite broad, we could only prove that efficient steering is possible in a limited case. For the class of follower algorithms in (2). we based the case on the observation that by far the most common no-regret algorithm for games in Hedge or multiplicative weights which is an instantiation of mirror descent with a negative entropy regularizer. Our results in this section are positive in the sense that it is possible to exploit learning agents in games under some (often realistic) assuption on their algorithm.
>
> Unfortunately in both of these cases the approach of learning the type and then using results from [Deng et al., 2019](https://arxiv.org/abs/1909.13861) is not straighforward. The follower's type is actually their payoff matrix, and learning the follower's mayoff matrix through interaction is  highly non-trivial--especially if one would like to do this efficiently and not wait for the follower to converge. Our solution is to make further assumptions on the type of algorithm the follower is using, and then use the strucure to help us learn the payoff matrix. Even this is nontrivial as can be seen in our results in Section 6. Furthermore, even if the optimizer has an approximation of the learner payoff matrix, since the method proposed in [Deng et al., 2019](https://arxiv.org/abs/1909.13861) is pessimistic by a unadjustable constant $\alpha$, as long as the learner's payoff structure is not recovered in an absolutely accurate way, directly applying the method in [Deng et al., 2019](https://arxiv.org/abs/1909.13861) would either be not pessimistic enough, such that the regret budget of the learner allows it to deviate from the Stackelberg equilibrium far enough to incur a huge Stackelberg regret, or be overly pessimistic, such that the pessimism itself brings a linear Stackelberg regret to the optimizer.
>
>
> ### Questions For Authors:
>
> > *"Can you elaborate on the contribution of non-sublinear Stackelberg regret in Section 6?"*
>
>
> Thank you for raising this question. We would like to emphasize that we are still assuming the learner to be no-regret in both parts in Section 6. More specifically, in Section 6.1, while Definition 6.1 does not imply an ascent algorithm being no-regret itself, we still need the no-regret property for the learner to drive it to an approximate best-response after having learned its paryoff structure. Meanwhile, many online ascent algorithms, e.g. online gradient ascent with proper step size, are themselves no-regret. In Section 6.2, stochastic mirror ascent with proper step size is no-regret as long as its regularizer is strongly convex. We will make a clear statement regarding this in our revision.

---

### Official Review · Reviewer_r9Wp · 2025-03-20

**Overall Recommendation:** 4

**Summary:**

The authors propose a new method to steer no-regret learners to a stackelberg equilibrium in repeated two player bimatrix games. There are two main contributions, The first is an impossibility result that there exists a no-regret learner that prevents an optimizer from achieving the stackelberg equilibrium when the learner's payoff matrix is unknown. However, with information about thel earner's  payoff matrix, the authors show that it is indeed possible to for the optimizer to steer the learner to the Stackelberg equilibrium.

Using the above two sufficient conditions, the paper argues that if we know that the learner is using an ascent algorithm, the optimiser can attain a sublinear regret by using a Binary Search algorithm (appendix E.1) to search through the simplex to estimate the pessimistic facets of the learner. Similarly, if the learner is known to use a mirror ascent algorithm with a known regulariser, it is possible to estimate the payoff matrix up to the equivalence class.

## update after rebuttal
After the inclusion of the experiments, I increase my score to a 4.

**Claims And Evidence:**

The claim for the impossibility result is convincing. The main result focuses on the idea of discovering the learner's payoff matrix by characterizing the learner's best response through facets of the learner's action polytope. The authors do take care to consider the pessimistic case, where if the learner is indifferent, it chooses the action that would be worse for the leader. The proofs appear to be rigorous and generally well-written. I couldn't find any obvious errors.

**Essential References Not Discussed:**

I would have expected some references to the machine learning communities work on opponent shaping

Lu, Christopher, et al. "Model-free opponent shaping." International Conference on Machine Learning. PMLR, 2022.

Foerster, Jakob N., et al. "Learning with opponent-learning awareness." arXiv preprint arXiv:1709.04326 (2017).

**Experimental Designs Or Analyses:**

There is no empirical evaluation (which I think is a major weakness).  It also seems unusual to have knowledge of the regularizer. Maybe the authors could comment more on this assumption.

**Methods And Evaluation Criteria:**

There doesn't seem to be much comparison to other methods. There are no experiments to showcase the power of this algorithm vs other benchmarks, even on simple games. While the theoretical results are thorough, it would help if there was more discussion on how these theoretical results compare to existing literature, or why other methods might not work in these cases.

**Other Comments Or Suggestions:**

n/a

**Other Strengths And Weaknesses:**

n/a

**Questions For Authors:**

I would be interested in the authors' comments on whether knowing the opponent's regularizer is a reasonable assumption, or if this can be relaxed. Maybe it is enough to know that they _are_ regularized, but not the exact quantity?

**Relation To Broader Scientific Literature:**

I think the paper did a good job of discussing related work. It is interesting to explore shaping, especially as AI systems become more ubiquitous. I would say, that while the theoretical results appear strong, it would strengthen the paper to include more big-picture results beyond _m = n = 2.

**Theoretical Claims:**

The theoretical claims appear to be valid. The impossibility result is proved with a counter-example. The requirement for disjoint pessimism seems reasonable (it maybe helpful to connect this to the non-existence of weak Stackelberg equilibria), and the extension to equivalences classes also seems correct.

---

> ### Author Rebuttal · Authors · 2025-04-01
>
> Before addressing the concerns, we would like to thank the reviewer for carefully reading the paper, providing insightful feedbacks and pointing out related empirical fields, all being really helpful for improving our work. Pleas find our response listed section-wise below:
>
> ### Methods and Evaluation Criteria:
> > *"There doesn't seem to be much comparison to other methods..."*
>
> To the best of our knowledge, our work is among the earliest to consider the problem of steering a no-regret learner to Stackelberg equilibrium *without* knowing its payoff structure. The only work close to this setting is [Brown et al., 2023](https://arxiv.org/pdf/2305.19496) but their method is only able to learn an approximate Stackelberg equilibrium efficiently when the learner is no-adaptive-regret, i.e. the learner regret between arbitrary time intervals should be bounded. Since we are considering a more general setting, establishing a fair comparison between our method and other methods can be challenging.
> ### Experimental Designs Or Analyses:
> > *"There is no empirical evaluation (which I think is a major weakness). It also seems unusual to have knowledge of the regularizer..."*
>
> We focus on the theoretical perspective of this problem and our main contribution is both the impossability result and---in light of this--- a few sufficient conditions under which no-Stackelberg regret is possible with provable upper bounds. We are happy to include empirical evaluation to illustrate the effectiveness of our result in the revision. Preliminary results seem to be in keeping with our theory. For the assumption of knowing the regularizer, please refer to **Question For Authors** part.
>
> ### Relation To Broader Scientific Literature:
> > *"... it would strengthen the paper to include more big-picture results beyond m = n = 2."*
>
> Indeed similar result holds for arbitrary $m$ as long as $n=2$. The idea is to separately treat each edge of the simplex as a special instance of the $2\times 2$ case. See Page 8, Line 428-439 and Appendix E.1 for more discussion. Please also refer to **Other Strengths And Weaknesses** section of our reply to reviewer abnn for the hardness when $n>2$.
>
> ### Essential References Not Discussed:
> > *"I would have expected some references to the machine learning communities work on opponent shaping..."*
>
> Thank you for pointing out these references. This is a relavant set of ideas that is related, though focuses on a different goal/situation.
>
> For the reference *"Foerster, Jakob N., et al. (2017)."*, they studied 2-player stochastic games with one learner provides conventional policy gradient while the other iterates in a one-step lookahead manner. The reference focus on *convergence* to equilibria, and assumes the learner has knowledge of the update rule, value functions and gradients of *both* agents *at the current time step*, while our work focus on *steering* the learner to *Stackelberg Equilibria* only assuming the optimizer knows the regularizer of the learner.
>
> For the reference *"Lu, Christopher, et al. 2022."*, which considered learning to play against other learning agents strategically, they formulated the meta games of partially observable stochastic games and designed a model-free algorithm that does meta-learning on the constructed MDP. Their work requires resetting the underlying POSG environment, while in our work we assume an online learning process where the optimizer minimizes its Stackelberg regret based on one single gameplay trajectory.
>
> ### Questions For Authors:
>
> > *"...whether knowing the opponent's regularizer is a reasonable assumption, or if this can be relaxed..."*
>
> This is a very interesting question and something we thought about but were unable to circumvent. Our results give sufficient conditions when steering a learning agent is possible (which in view of our impossibility result is a hard problem in general). To do so we decided to focus on classes of algorithms for the follower that were commonly analyzed and used for learning in matrix games. Towards this we identified that most common algorithms for matrix games are based around the idea of multiplicative weights (or Hedge/exponentiated gradients) which is the result of using the negative entropy as a regularizer. The second other common class is projected gradient descent which results from the use of the $L_2$ Euclidean norm. Against both of these algorithms we showed that efficient steering is indeed possible.
>
> The requirement of knowing the geometry in which the follower optimizes is strong but seems to be crucial for efficiently learning to steer them, since it allows the leader to glean information about $B$ from observation of their consecutive actions (and not have to wait for the follower to converge as in other papers). We believe it is a very interesting open problem whether the follower using mirror descent with arbitrary regularizers inadvertently reveals information about their payoff B in such a fine grained manner.

---

> > ### Comment · Reviewer_r9Wp · 2025-04-01
> >
> > Thank you for your response. I am certainly willing to increase my score with the inclusion of the empirical results.

---

> > > ### Author Response · Authors · 2025-04-05
> > >
> > > Thank you for you follow-up comment. We provide empirical simulations for Algorithm 1 and Algorithm 4 proposed in Section 6 and plot the learning and payoff dynamics in the following [Link](https://anonymous.4open.science/r/Learning_to_Steer_Learners_in_Games-28F5). Below we summarize our simulation results:
> > >
> > > ### Empirical Simulation for Section 6.1
> > >
> > > For all experiments in this Section, we assume the learner is using Online Gradient descent (OGD) with step size $\eta_t=\frac{\eta_0}{\sqrt{t}}$. For the purpose of properly displaying the interaction and learning process, we choose different $\eta_0$ for different game instances. For each game instance, we compare the performance and learning dynamics for optimizer algorithm being either OGD or Binary Search explore-then-commit (BS). For Binary Search, we set the accuracy margin $d=0.01$. For each game instance, we plot both the payoff and the strategy (indicated by its 0-th entry) of each player at different time steps. We assume optimizer is the row player and learner is the column player.
> > >
> > > *Matching Pennies:*
> > >
> > > We first test repeated Matching pennies, where the payoff matrix is given by:
> > >
> > > | Payoff (Optimizer, Learner) | H | T |
> > > | -------------- | -- | -- |
> > > | H               | (1, -1)    | (-1, 1)    |
> > > | T                | (-1, 1)    | (1, -1)    |
> > >
> > > The unique Nash equilibrium and the Stackelberg equilibria of this game all have $x=(1/2,1/2)^T$.
> > >
> > > We obtain the curve in OGD_vs_BS_mp.pdf. We can see that when both players are using OGD, the trajectory keeps oscillating and does not converge to the Nash equilibrium. In comparison, when the optimizer uses BS, it quickly learns its real underlying Stackelberg equilibrium (which is also the Nash) and commits to it, yielding a stable learning dynamics.
> > >
> > > *Constructed Game Instance 1:*
> > >
> > > Below we show that BS indeed yields a smaller Stackelberg regret than OGD. We construct the following instance:
> > >
> > > | Payoff (Optimizer, Learner) | L | R |
> > > | ----- | ----- | --- |
> > > | U                | (5, -2)    | (0, 2)     |
> > > | D                | (0, 3)     | (3, -3)    |
> > >
> > > The unique Stackelberg equilibrium action for the optimizer is $x=(3/5, 2/5)^T$ with Stackelberg value $3$. The learning dynamics is shown in OGD_vs_BS_plot1.pdf or OGD_vs_BS_plot1_sep.pdf for separately plotted curves.
> > >
> > > We notice again that when the optimizer is using OGD, the algorithm fails to converge. Also, after the optimizer commits to the pessimistic Stackelberg solution, the learner slowly converges to the best response induced by the Stackelberg equilibrium and steers the optimizer payoff close to the Stackelberg value, which is higher on average than the payoff using SGD.
> > >
> > > *Constructed Game Instance 2:*
> > >
> > > Below we construct a game instance that has a unique Nash equilibrium to which OGD converges, and a unique Stackelberg equilibrium with higher optimizer utility than that of Nash:
> > >
> > > | Payoff (Optimizer, Learner) | L | R |
> > > | --------- | --- | ---- |
> > > | U                | (2, 1)     | (0, 0)     |
> > > | D                | (3, 0)     | (1, 2)     |
> > >
> > > The unique Nash equilibrium is $x=y=(0,1)^T$, while the unique Stackelberg equilibrium is $x=(2/3, 1/3)^T, y=(1,0)^T$. The optimizer payoff at Nash is $1$, while its Stackelberg value is $2$. The simulation result is shown in OGD_vs_BS_plot2.pdf. The plot above shows that even if both converges, BS and OGD converge to different equilibria, while BS yields a higher average payoff.
> > >
> > > ### Empirical Simulation for Section 6.2
> > >
> > > In this section we show the effectiveness of Algorithm 4 and illustrate the necessity of *pessimism*. Here we assume the optimizer is using Algorithm 2, but with different pessimism levels $d\in\{0.01, 0.02, 0.05\}$. We assume that the learner is using Stochastic Mirror descent with KL regularizer. For each pure strategy of the optimizer, we set the number of steps for exploration to be $k=50$. We consider the following game instance:
> > >
> > > | Payoff (Optimizer, Learner) | L | R |
> > > | ---------------- | ---- | ------ |
> > > | U                | (0, 2)     | (1, -2)    |
> > > | D                | (5, -3)    | (0, 3)     |
> > >
> > > The unique Stackelberg equilibrium of this game is $x=(3/5, 2/5)^T$ with optimizer payoff $2$.
> > >
> > > We plot the payoffs and strategies of both player at each time step with different $d$ in KLestimation_plot1.pdf. We can see that for larger $d$, the optimizer is being more pessimistic and chooses an action that is farther away from the Stackelberg equilibrium. Although leading to a lower Stackelberg value, for the less pessimistic choices of $d$, since the committed optimizer strategy $\tilde{x}$ is too close to the Stackelberg equilibrium where the learner is indifferent from all mixed strategies, the gradients of the learner payoff will be extremely small and thus takes a lot longer to converge, leading to a lower payoff before convergence.
> > >
> > > We appreciate further discussions and are happy to address any additional questions you may have.
> > >
> > > Thank you again for your efforts and suggestions,
> > >
> > > Authors

---

### Decision · Program_Chairs · 2025-05-01

**Decision:**

Accept (poster)

**Comment:**

After somewhat extensive discussion with the authors, all reviewers agree the paper should be accepted.  I'd encourage the authors to consider the various feedback and discussions as they create the final version of the paper.